# $CO_2$ emissions from peat-draining rivers regulated by water $p$H

Alexandra Klemme[1], Tim Rixen[2,3], Denise Müller-Dum[1], Moritz Müller[4], Justus Notholt[1], and
Thorsten Warneke[1]

[1]Institute of Environmental Physics, University of Bremen, Otto-Hahn-Allee 1, 28359 Bremen, Germany
[2]Leibniz Center for Tropical Marine Research, Fahrenheitstr. 6, 28359 Bremen, Germany
[3]Institute of Geology, University of Hamburg, Bundesstr. 55, 20146 Hamburg, Germany
[4]Faculty of Engineering, Computing, and Science, Swinburne University of Technology Sarawak Campus, Jalan Simpang
Tiga, 93350 Kuching, Sarawak, Malaysia

**Correspondence:** Alexandra Klemme (aklemme@uni-bremen.de)

**Abstract.** Southeast Asian peatlands represent a globally significant carbon store that is destabilized by land-use changes like
deforestation and the conversion into plantations, causing high carbon dioxide ($CO_2$) emissions from peat soils and increased
leaching of peat carbon into rivers. While this high carbon leaching and consequentially high DOC concentrations suggest that
$CO_2$ emissions from peat-draining rivers would be high, estimates based on field data suggest they are only moderate. In this

study, we offer an explanation for this phenomenon by showing that carbon decomposition is hampered by the low $p$H in peat-
draining rivers. This limits $CO_2$ production in and emissions from these rivers. We find an exponential $p$H limitation that shows
good agreement with laboratory measurements from high latitude peat soils. Additionally, our results suggest that enhanced
input of carbonate minerals increases $CO_2$ emissions from peat-draining rivers by counteracting the $p$H limitation. As such
inputs of carbonate minerals can occur due to human activities like deforestation of river catchments, liming in plantations, and

enhanced weathering application, our study points out an important feedback mechanism of those practices.

## 1 Introduction

Rivers and streams emit high amounts of carbon dioxide ($CO_2$) to the atmosphere (Cole et al., 2007), but estimates of these
emissions ($0.6-1.8\,\mathrm{PgC\,yr^{-1}}$) are highly uncertain (Aufdenkampe et al., 2011; Raymond et al., 2013). Studies agree that more
than three-quarters of global river $CO_2$ emissions occur in the tropics (Raymond et al., 2013; Lauerwald et al., 2015). River $CO_2$

emissions are controlled by the partial pressure difference between $CO_2$ in the atmosphere and the river water (Raymond et al.,
2012). Riverine $CO_2$ is fed by the decomposition of organic matter leached from soils (Wit et al., 2015), by direct leaching of
dissolved $CO_2$ from soil respiration (Abril and Borges, 2019; Lauerwald et al., 2020), and by photomineralization of dissolved
organic carbon (DOC) (Nichols and Martin, 2021; Zhou et al., 2021). Studies suggest Southeast Asia as a potential hotspot for
river $CO_2$ emissions (Raymond et al., 2013; Lauerwald et al., 2015) due to the presence and degradation of carbon-rich peat

soils. However, measurements of river $CO_2$ emissions from this region are sparse.

More than half of the known tropical peatlands are located in Southeast Asia (Page et al., 2011; Dargie et al., 2017), whereby
$84\%$ of these are Indonesian peatlands, mainly on the islands of Sumatra, Borneo, and Irian Jaya (Page et al., 2011). Already
in 2010, land-use change affected $90\%$ of the peatlands located on Sumatra and Borneo (Miettinen and Liew, 2010) and turned

them from $CO_2$ sinks to $CO_2$ sources (Hooijer et al., 2010; Miettinen et al., 2017; Hoyt et al., 2020). Enhanced decomposition

in disturbed peatlands additionally increases the leaching of organic matter from soils into peat-draining rivers (Moore et al., 2013; Rixen et al., 2016; Cook et al., 2018). According to Regnier et al. (2013), land-use change remobilizes $(1.0 \pm 0.5)\,\mathrm{Pg}$ of soil organic carbon per year of which $40\,\%$ are decomposed in rivers and emitted as $CO_2$ to the atmosphere. The resulting $CO_2$ emissions of $0.4\,\mathrm{PgC\,yr^{-1}}$ represent $33\,\%$ of the total $CO_2$ emissions from rivers (Regnier et al., 2013).

Peat soils are rich in carbon, causing high concentrations of DOC in peat-draining rivers that increase with increasing peat

coverage of the river catchments (Wit et al., 2015). However, despite high carbon leaching rates that cause DOC concentrations which can be more than four times higher than those in temperate regions (Butman and Raymond, 2011; Müller et al., 2015; Gandois et al., 2020), measured $CO_2$ fluxes from tropical rivers with high peat coverage ($18 - 41\,\mathrm{gC\,m^{-2}\,yr^{-1}}$) hardly exceed those measured for rivers in temperate regions ($18.5\,\mathrm{gC\,m^{-2}\,yr^{-1}}$, Müller et al., 2015; Butman and Raymond, 2011). Different reasons for this were suggested in literature. Müller et al. (2015) suggested short residence times of peat derived DOC in rivers

due to the location of peatlands near the coast as a possible cause. Other suggestions are the recalcitrant nature of DOC (Müller et al., 2016) and the lack of oxygen ($O_2$, Wit et al., 2015) which both lower the rate of DOC decomposition. Moreover, Borges et al. (2015) suggested a limitation of bacterial production and the resulting DOC decomposition in African peat-draining rivers as a consequence of low $p$H based on observations at rivers in the Congo basin.

The assumption of low $O_2$ concentrations and $p$H as cause for moderate $CO_2$ emissions is supported by the regulating effect of

these parameters on decomposition rates in peat soils. $p$H and $O_2$ are the key parameters that limit the activity of the decomposition impelling enzyme phenol oxidase (Pind et al., 1994; Freeman et al., 2001). Phenol oxidase is needed to decompose phenolic compounds that are especially present in tropical peat soils (Hodgkins et al., 2018; Yule et al., 2018). Those phenolic compounds are more rapidly decomposed in the upper layer of peat soils than in deep peat (Gandois et al., 2014). Studies agree that the limiting effect of oxygen on decomposition is accurately represented by the Michaelis-Menten kinetics (Fang

and Moncrieff, 1999; Pereira et al., 2017). This approach assumes that DOC decomposition is linearly limited for low $O_2$ concentrations but that there is no limitation for higher $O_2$ concentrations once they are sufficient to meet the decomposition demands (Keiluweit et al., 2016). Due to high rates of decomposition caused by the carbon-rich environment and low rates of photosynthesis caused by low nutrient concentrations and dark water colours that limit light availability to algae, peat-draining rivers are usually undersaturated with regard to atmospheric $O_2$ (Wit et al., 2015; Baum and Rixen, 2014). Still, their $O_2$ con-

centrations exceed those in peat soils due to gas exchange with the atmosphere (Müller et al., 2015; Rixen et al., 2008) and thus are assumed to limit decomposition less strongly than in peat soils (Pind et al., 1994). The same applies to the $p$H limitation, as $p$H in peat-draining rivers is usually higher than in peat soils (Pind et al., 1994). Other than for $O_2$ limitation, however, the form of the $p$H limitation is still subject to discussion. Linear (Sinsabaugh, 2010) and exponential (Williams et al., 2000; Kang et al., 2018) correlations have been stated in the literature.

This study aims at quantifying the limiting impact of $p$H and $O_2$ on the DOC decomposition in peat-draining rivers to explain the moderate $CO_2$ emissions observed from these rivers. We analysed data from ten Southeast Asian peat-draining rivers with DOC concentrations between $200\,\mathrm{\mu mol\,L^{-1}}$ and $3{,}000\,\mathrm{\mu mol\,L^{-1}}$ and $p$H and $O_2$ concentrations ranging from 3.8 to 7.1 and from $50\,\mathrm{\mu mol\,L^{-1}}$ to $200\,\mathrm{\mu mol\,L^{-1}}$, respectively.

## 2 Materials and methods

This study's methods were separated into two parts. The first part provides information on the study area, conducted measurement campaigns and collected data that our analyses are based on. The second part describes the processes and equations used to quantify the decomposition dependency on $O_2$ and $pH$.

### 2.1 Measurement campaigns and study area

#### 2.1.1 Study area

Southeast Asian peatlands store $42\,\mathrm{Pg}$ soil carbon across an area of $271{,}000\,\mathrm{km}^2$ (Hooijer et al., 2010). More than $97\%$ of these peat soils are located in lowlands (Hooijer et al., 2006). The development of peatlands in Southeast Asia is favoured by its tropical climate with high precipitation rates that range between $120\,\mathrm{mm}$ in July and $310\,\mathrm{mm}$ in November with an annual mean of $2{,}700\,\mathrm{mm\,yr}^{-1}$ (Yatagai et al., 2020). Due to land-use changes like deforestation and the conversion into plantations, today less than one-third of those Southeast Asian peatlands remain covered by peat swamp forests, while in 1990 it were more than three-quarters (Miettinen et al., 2016). Southeast Asian rivers mostly originate in mountain regions and cut through coastal peatlands on their way to the ocean (Fig. 1). Measurement data included in this study were obtained in river parts that flow through peat soils to capture the influence of peatlands on the carbon dynamics in the rivers. The impact of sampling locations and seasonality are discussed in the appendix B.

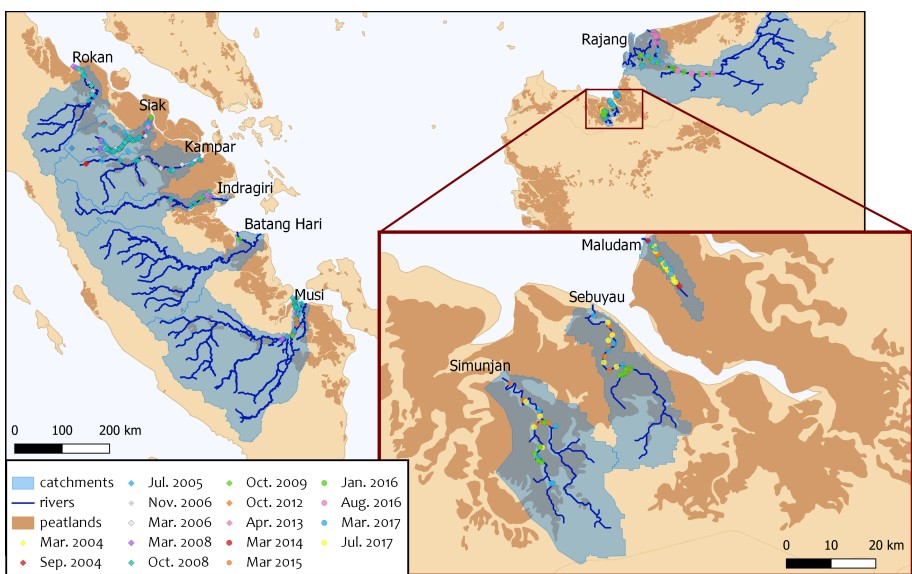

**Figure 1.** Map of river catchments with the location of peat areas. Blue lines indicate the main rivers. Blue shaded areas outline the river basins and brown areas indicate peatlands. Coloured data points indicate the sampling stations of the individual campaigns

The collective data for this study were derived from four rivers on Borneo (Sarawak, Malaysia) and six rivers on Sumatra (Indonesia). The investigated rivers on Borneo are the Rajang, Simunjan, Sebuyau, and Maludam and the rivers surveyed on Sumatra are the Rokan, Kampar, Indragiri, Batang Hari, Musi, and Siak (Fig. 1). We additionally include data from the Siak's tributaries Tapung Kiri, Tapung Kanan and Mandau. River peat coverages range from $4\%$ in the Musi catchment to $91\%$ in the Maludam catchment, whereby the bigger rivers that originate in the uplands generally have lower peat coverages than smaller coastal rivers.

### 2.1.2 River expeditions and measured parameters

Data were derived from a total of 16 campaigns in Sumatra and Sarawak (Fig. 1, Tab. A1). For the Indonesian rivers, ten measurement campaigns between 2004 and 2013 were conducted. We use published data from Baum et al. (2007) for the Mandau, Tapung Kanan and Tapung Kiri rivers, from Wit et al. (2015) for the Siak, Indragiri, Batang Hari and Musi rivers and from Rixen et al. (2016) for the Rokan and Kampar rivers. $CO_2$ measurements are available for the campaigns performed after 2008.

For the Malaysian rivers, measurements were performed in six campaigns between 2014 and 2017. We use data published by Müller-Dum et al. (2019) and Martin et al. (2018) for the Rajang river and by Müller et al. (2015) for the Maludam campaigns in 2014 and 2015. Additional campaigns for this study were conducted in March 2015 at the Simunjan and Sebuyau rivers as well as in January 2016, March 2017 and July 2017 at the Simunjan, Sebuyau and Maludam rivers. Measurements of DOC, $CO_2$ and $O_2$ concentrations as well as water $p$H, water temperatures ($T$) and gas exchange coefficients ($k_{600}$) for these additional campaigns were performed in the same manner as during the 2014 Maludam campaign (Müller et al., 2015). However, due to instrumental problems, the $CO_2$, $O_2$ and $p$H data measured at the Simunjan river in 2016 were not available for our analysis. Table 3 lists the averaged river parameters, including the catchments' peat coverages and atmospheric $CO_2$ fluxes.

During the January 2016, March 2017 and July 2017 campaigns, concentrations of particulate inorganic carbon (PIC) in form of $CaCO_3$ were measured in addition to the other parameters. These data are not included in the before-mentioned studies. Therefore, we describe the measurement principle here. Discrete water samples, taken from approximately $1\,\mathrm{m}$ below the water surface, were filtered through pre-weighed and pre-combusted glass fiber filters ($0.7\,\mu\mathrm{m}$) to sample particulate material within the water volume. To determine the particulate carbon (organic and inorganic), the samples were catalytically combusted at $1,050\,^{\circ}\mathrm{C}$ and combustion products were measured by thermal conductivity using an Euro EA3000 Elemental Analyzer. The PIC was determined from the difference between this total particulate carbon and particulate organic carbon that was measured after addition of 1 molar hydrochloric acid in order to remove the inorganic carbon from the sample.

### 2.1.3 River catchment size and peat coverage

River catchment sizes were derived from Hydro-SHEDS (Lehner et al., 2006) at $15\,\mathrm{s}$ resolution in WGS 1984 Web Mercator Projection. Sub-basins belonging to the catchments were identified using the HydroSHEDS $15\,\mathrm{s}$ flow directions data set and added to the main basins. The estimated accuracy of final catchment lines is $0.4\%$.

Catchment peat coverage was derived from peat maps downloaded from www.globalforestwatch.org for Indonesia and Malaysia. The Indonesian peatland map was published by the Ministry of Agriculture in 2012. The Malaysian peat map was made available by Wetlands International in 2004 and is based on a national inventory by the Land and Survey Department of Sarawak (1968). Both maps include peatlands in different conditions, from undisturbed peat swamp forest to disturbed peat under plantations, which is nowadays widespread in those countries. Peat coverage was determined from the areal extent of peatlands in the catchment divided by catchment size. Peat coverages derived using other peat maps are compared in appendix C.

## 2.2 Analysis of decomposition dependency on $pH$ and $O_2$

Decomposition dependencies on $pH$ and $O_2$ were derived based on the assumption that concentrations of DIC and $O_2$ in peat-draining rivers, as a first approximation, are derived from an equilibrium between gas exchange with the atmosphere and DOC decomposition in the river water. This approximation assumes photosynthetic $CO_2$ consumption, photomineralization, and direct $CO_2$ leaching from soils to be negligible. We discuss the impact of these processes later on. In this section, we introduce the calculation of atmospheric gas exchange fluxes and decomposition rates. Then we derive equations to quantify decomposition limitations by $pH$ and $O_2$ based on an equilibrium between these two processes.

### 2.2.1 Gas exchange between rivers and the atmosphere

Atmospheric $CO_2$ fluxes from rivers were calculated from $CO_2$ gas exchange coefficients and river $CO_2$ concentrations according to

$$F_{CO_2} = k_{CO_2}(T) \cdot (CO_2 - K_{CO_2}(T) \cdot pCO_2^a). \tag{1}$$

Exchange coefficients for $CO_2$ ($k_{CO_2}(T)$) were calculated from $k_{600}$ and water temperature according to Wanninkhof (1992) as

$$k_{CO_2} = k_{600} \cdot \left( \frac{1911.1 - 118.11 \cdot T + 3.4527 \cdot T^2 - 0.041320 \cdot T^3}{600} \right)^{-n}. \tag{2}$$

An exponent of $n = 1/2$ (valid for rough surfaces; Zappa et al., 2007) was used for the rivers. The temperature $T$ is given in °C. $pCO_2^a$ is the atmospheric partial pressure of $CO_2$ ($\approx 400\,\mu atm$) and $K_{CO_2}$ describes the temperature dependent Henry coefficient for $CO_2$, which was calculated according to Weiss (1974) as

$$\ln K_{CO_2} = -58.0931 + 90.5069 \cdot \frac{100}{T} + 22.2940 \cdot \ln \left( \frac{T}{100} \right). \tag{3}$$

$k_{CO_2}$ and $K_{CO_2}$ derived for the individual rivers are listed in the appendix in Tab. A2.

Atmospheric $O_2$ fluxes ($F_{O_2}$) were derived analogously to $F_{CO_2}$ with $k_{O_2}(T)$ calculated according to Wanninkhof (1992) as

$$k_{O_2} = k_{600} \cdot \left( \frac{1800.6 - 120.10 \cdot T + 3.7818 \cdot T^2 - 0.047608 \cdot T^3}{600} \right)^{-n} \tag{4}$$

and Henry coefficients for $O_2$ ($K_{O_2}$) calculated according to Weiss (1970) as

$$\ln K_{O_2} = -58.3877 + 85.8079 \cdot \frac{100}{T} + 23.8439 \cdot \ln \frac{T}{100}. \tag{5}$$

$k_{O_2}$ and $K_{O_2}$ for the individual rivers in this study are also listed in the appendix in Tab. A2.

### 2.2.2 Decomposition rates and their dependency on $p$H and $O_2$

The decomposition rate of DOC ($R$) is defined as molecules of $CO_2$ that are produced per available molecules of DOC during a specific time step and thus represents the proportionality factor between the $CO_2$ production rate and the DOC concentration:

$$R = \frac{\Delta CO_2}{DOC \cdot \Delta t} \quad \Rightarrow \quad \frac{\partial CO_2}{\partial t} = R \cdot DOC. \tag{6}$$

As discussed before, $R$ can be limited by $O_2$ concentrations and by $p$H. We use an $O_2$ limitation factor that is based on the Michaelis-Menten equation ($L_{O_2} = \frac{O_2}{K_m + O_2}$) as suggested by Pereira et al. (2017). For $p$H limitation, we consider an exponential limitation factor ($L_{pH}^{\exp} = \exp\left(\lambda \cdot (pH - pH_0)\right)$) as suggested by Williams et al. (2000) and a linear limitation factor ($L_{pH}^{\lin} = \frac{pH}{pH_0}$) as suggested by Sinsabaugh (2010). Considering the definition of $p$H as negative decadic logarithm of $H^+$ activity ($\{H^+\}$), the exponential limitation factor is equivalent to a linear correlation with $\{H^+\}^{\frac{\lambda}{\ln(10)}}$.

The $CO_2$ production rates due to DOC decomposition for the linear and the exponential $p$H limitation approach are thus defined as:

$$\begin{aligned}
\left(\frac{\partial CO_2}{\partial t}\right)_{\lin} &= R_{\max} \cdot L_{O_2} \cdot L_{pH}^{\lin} \cdot DOC = R_{\max} \cdot \frac{O_2}{K_m + O_2} \cdot \frac{pH}{pH_0} \cdot DOC \\
\left(\frac{\partial CO_2}{\partial t}\right)_{\exp} &= R_{\max} \cdot L_{O_2} \cdot L_{pH}^{\exp} \cdot DOC = R_{\max} \cdot \frac{O_2}{K_m + O_2} \cdot \exp\left(\lambda \cdot (pH - pH_0)\right) \cdot DOC.
\end{aligned} \tag{7}$$

$R_{\max}$ is the maximum decomposition rate. $K_m$ is the Michaelis constant for $O_2$ inhibition. It is also called the half saturation constant and gives the $O_2$ concentration at which $O_2$ limits decomposition by $50\%$ (Loucks and Beek, 2017). $\lambda$ is the exponential $p$H inhibition constant and $pH_0$ is a normalization constant that was set to 7.5 since this is reported to be the optimal $p$H for the activity of the decomposition impelling enzyme phenol oxidase (Pind et al., 1994; Kocabas et al., 2008). Calculations of $pH_0$ based on our data and the exponential $p$H approach are described in the appendix D4. They yield an optimum $p$H of approximately 7.2 and thus agree well with the $pH_0$ of 7.5 used in this study.

$L_{O_2}$ and $L_{pH}$ can take values between 0 and 1. Thus, Eq (7) is only valid for $pH \leqslant pH_0$. For higher water $p$H, a different approach would be needed. However, for the rivers in this study Eq. (7) is sufficient since their $p$H is $< 7.5$ (Tab. 3). The limitation factors represent the fraction of decomposition that is remaining after the limitation by the parameter. Later on, we refer to the fraction by which decomposition is limited, which is $(1 - L_{pH})$ for $p$H limitation and $(1 - L_{O_1})$ for $O_2$ limitation. The total fraction by which $p$H and $O_2$ limit decomposition is given by $(1 - L_{pH} \cdot L_{O_2})$. When $O_2$ concentrations and water $p$H are high enough not to limit the decomposition rate, Eq. (7) simplifies to Eq. (6) with $R = R_{\max}$.

### 2.2.3 Least-squares optimization to quantify the $p$H and $O_2$ impact on decomposition rates

As mentioned before, we base our calculations on the assumption that DIC concentrations in peat-draining rivers, result from an equilibrium between $CO_2$ emissions and $CO_2$ production by decomposition. Thus, we optimized the parameters in Eq. (7) such that the production of $CO_2$ in the water volume beneath a specific surface area equals the atmospheric $CO_2$ flux through

this area. The $CO_2$ production is calculated by multiplication of Eq. (7) with the product of river depth $d$ and surface area $A$ and the $CO_2$ emissions are calculated by multiplication of Eq. (1) with the surface area $A$:

$$d \cdot A \cdot R_{\max} \cdot L_{O_2} \cdot L_{pH} \cdot \text{DOC} = A \cdot k_{CO_2}(T) \cdot (CO_2 - K_{CO_2}(T) \cdot pCO_2^{\mathrm{a}}). \tag{8}$$

Analogously, river $O_2$ concentrations result from an equilibrium between the atmospheric $O_2$ flux and $O_2$ consumption due to decomposition. During decomposition, the $O_2$ consumption is proportional to the $CO_2$ production ($\Delta O_2 = -b \cdot \Delta CO_2$). The proportionality factor $b$ is usually $< 1$ since a fraction of the $O_2$ used for decomposition is taken from the oxygen content in the dissolved organic matter (Rixen et al., 2008). Thus, the equilibrium between $O_2$ consumption within the water volume and $O_2$ flux through the surface area can be written as

$$-b \cdot d \cdot A \cdot R_{\max} \cdot L_{O_2} \cdot L_{pH} \cdot \text{DOC} = A \cdot k_{O_2}(T) \cdot (O_2 - K_{O_2}(T) \cdot pO_2^{\mathrm{a}}). \tag{9}$$

In order to compare these dependencies to measured data, Eq. (8) and Eq. (9) were analytically solved for $CO_2$ and for $O_2$, respectively. The resulting equations based on linear $pH$ limitation ($L_{pH}^{\mathrm{lin}} = \frac{pH}{pH_0}$) are listed in Tab. 1. The analogously derived equations for $CO_2$ and $O_2$ based on the exponential $pH$ approach ($L_{pH}^{\mathrm{exp}} = \exp(\lambda \cdot (pH - pH_0))$) are listed in Tab. 2.

Based on these equations, least-squares optimizations were performed to derive the decomposition parameters $R_{\max}$, $b$, $K_m$ and $\lambda$ such that $CO_2(\text{DOC}, pH, O_2, T)$ and $O_2(\text{DOC}, pH, T)$ are simultaneously optimized for the measured parameters of DOC, $pH$, $T$, $CO_2$ and $O_2$.

**Table 1.** Equations to derive $CO_2$ and $O_2$ based on the linear $pH$ approach.

| | |
|---|---|
| $\mathbf{CO_2}(\text{DOC}, pH, O_2, T) =$ | $K_{CO_2}(T) \cdot pCO_2^{\mathrm{a}} + \dfrac{d \cdot R_{\max} \cdot \text{DOC} \cdot \frac{O_2}{K_m + O_2} \cdot \frac{pH}{pH_0}}{k_{CO_2}(T)}$ |
| $\mathbf{O_2}(\text{DOC}, pH, T) =$ | $\sqrt{\left(\dfrac{b \cdot d \cdot R_{\max} \cdot \text{DOC} \cdot \frac{pH}{pH_0} + k_{O_2}(T) \cdot (K_m - K_{O_2}(T) \cdot pO_2^{\mathrm{a}})}{2 \cdot k_{O_2}(T)}\right)^2 + K_{O_2}(T) \cdot pO_2^{\mathrm{a}} \cdot K_m} - \dfrac{b \cdot d \cdot R_{\max} \cdot \text{DOC} \cdot \frac{pH}{pH_0} + k_{O_2} \cdot (K_m - K_{O_2}(T) \cdot pO_2^{\mathrm{a}})}{2 \cdot k_{O_2}(T)}$ |

Equations to derive $CO_2$ from measured temperature ($T$), DOC, $pH$ and $O_2$ as well as to derive $O_2$ from measured $T$, DOC and $pH$. $R_{\max}$, $K_m$ and $b$, derived via least-squares optimization using measured DOC, $pH$, $T$, $O_2$ and $CO_2$ data of the investigated rivers, are listed in Tab. 5.

**Table 2.** Equations to derive $CO_2$ and $O_2$ based on the exponential $pH$ approach.

| | |
|---|---|
| $\mathbf{CO_2}(\text{DOC}, pH, O_2, T) =$ | $K_{CO_2}(T) \cdot pCO_2^{\mathrm{a}} + \dfrac{d \cdot R_{\max} \cdot \text{DOC} \cdot \frac{O_2}{K_m + O_2} \cdot \exp(\lambda \cdot (pH - pH_0))}{k_{CO_2}(T)}$ |
| $\mathbf{O_2}(\text{DOC}, pH, T) =$ | $\sqrt{\left(\dfrac{b \cdot d \cdot R_{\max} \cdot \text{DOC} \cdot \exp(\lambda \cdot (pH - pH_0)) + k_{O_2}(T) \cdot (K_m - K_{O_2}(T) \cdot pO_2^{\mathrm{a}})}{2 \cdot k_{O_2}(T)}\right)^2 + K_{O_2}(T) \cdot pO_2^{\mathrm{a}} \cdot K_m} - \dfrac{b \cdot d \cdot R_{\max} \cdot \text{DOC} \cdot \exp(\lambda \cdot (pH - pH_0)) + k_{O_2} \cdot (K_m - K_{O_2}(T) \cdot pO_2^{\mathrm{a}})}{2 \cdot k_{O_2}(T)}$ |

Equations to derive $CO_2$ from measured temperature ($T$), DOC, $pH$ and $O_2$ as well as to derive $O_2$ from measured $T$, DOC and $pH$. $R_{\max}$, $K_m$, $\lambda$ and $b$, derived via least-squares optimization using measured DOC, $pH$, $T$, $O_2$ and $CO_2$ data of the investigated rivers and are listed in Tab. 5.

The equations in Tab. 1 and Tab. 2 depend on the river gas exchange coefficients for $CO_2$ ($k_{CO_2}$) and $O_2$ ($k_{O_2}$), which both depend on $k_{600}$. Those exchange coefficients are poorly constrained and spatially as well as temporally extremely variable. The $k_{600}$ we list in this study are based on a variety of techniques, including floating chamber measurements (Müller et al., 2015),

calculations based on wind speed and catchment parameters (Müller-Dum et al., 2019) and balance models of water parameters (Rixen et al., 2008). Although all of those estimates remain highly uncertain, we find a fairly good agreement between $k_{600}$ and river depths ($d$, Fig. A1). We therefore use a fixed ratio of $k_{600}/d = (7.0 \pm 0.5) \cdot 10^{-6}\,\mathrm{s}^{-1}$ for the least-squares optimizations rather than individual exchange coefficients and depths of the rivers.

## 3 Results

### 3.1 Correlation with peat coverage

The data presented in Tab. 3 yield a linear increase of river DOC concentration with peat coverage (Fig. 2a) as well as a negative linear correlation between river $p$H and peat coverage (Fig. 2b). The river $CO_2$ concentration shows a strong increase for peat coverages $< 30\%$. Despite further increase in DOC concentrations, $CO_2$ concentrations in rivers with peat coverage $> 30\%$ level off, resulting in a fairly constant $CO_2$ for peat coverages $> 50\%$ (Fig. 2c). The river $O_2$ shows an opposite behaviour to the $CO_2$. $O_2$ concentrations initially decrease with increasing peat coverage and show a decline in the regression rate for high peat coverages, resulting in a minimum $O_2$ concentration of approximately $65\,\mu\mathrm{mol\,L}^{-1}$ (Fig. 2d).

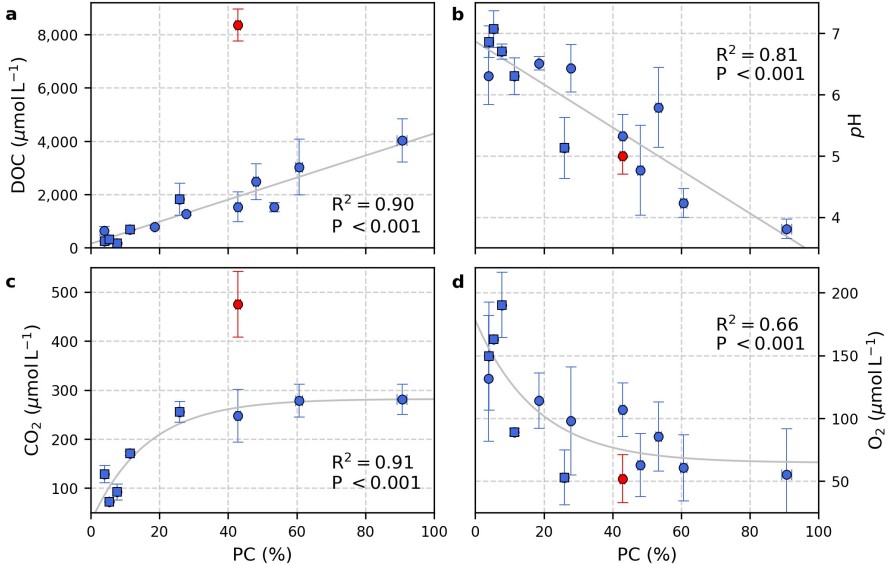

**Figure 2.** Correlation of peat coverage (PC) with (a) DOC, (b) $p$H, (c) $CO_2$ & (d) $O_2$. Each data point represents one river. Variability is indicated by the error bars, which are given by standard deviation. For the Simunjan river, the January 2016 and March 2017 campaigns (Simunjan$_2$, see Tab. 3 and Tab. 4), indicated by red data points, were separated from the other Simunjan campaigns (Simunjan$_1$) and excluded from the correlations due to strong deviations from the other campaigns that imply an additional process discussed in Sect. 4.4. Ordinary least-squares optimizations were used to calculate linear correlations with DOC and $p$H and exponential correlations with $CO_2$ and $O_2$. Rivers included in a previous study investigating these correlations (Wit et al., 2015) are indicated by squares.

**Table 3.** Measured data from the investigated rivers.

| River | peat coverage (%) | $p$H | $T$ (°C) | DOC ($\mu$mol L$^{-1}$) | $O_2$ ($\mu$mol L$^{-1}$) | $CO_2$ ($\mu$mol L$^{-1}$) | $k_{600}$ (cm h$^{-1}$) | $F_{CO_2}$ (gC m$^{-2}$ d$^{-1}$) |
|---|---|---|---|---|---|---|---|---|
| Musi | $4.0 \pm 0.1$ | $6.9 \pm 0.3$ | $30.6 \pm 0.3$ | $244 \pm 5$ | $149 \pm 43$ | $128 \pm 18$ | $17 \pm 4$ | $2.8 \pm 2.9$ |
| Batang Hari | $5.4 \pm 0.1$ | $7.1 \pm 0.3$ | $30.0 \pm 0.1$ | $321 \pm 4$ | $163 \pm 20$ | $72 \pm 20$ | $17 \pm 4$ | $1.4 \pm 0.4$ |
| Indragiri | $11.4 \pm 0.2$ | $6.3 \pm 0.3$ | $31.5 \pm 0.1$ | $692 \pm 5$ | $89 \pm 20$ | $171 \pm 20$ | $17 \pm 4$ | $3.8 \pm 1.2$ |
| Siak | $25.9 \pm 0.4$ | $5.1 \pm 0.5$ | $30.0 \pm 0.2$ | $1,829 \pm 601$ | $53 \pm 22$ | $256 \pm 21$ | $17 \pm 4$ | $5.9 \pm 2.6$ |
| Kampar | $27.8 \pm 0.5$ | $6.4 \pm 0.4$ | $29.4 \pm 0.7$ | $1,280 \pm 44$ | $98 \pm 43$ | n.d. | n.d. | n.d. |
| Rokan | $18.6 \pm 0.3$ | $6.5 \pm 0.1$ | $28.9 \pm 1.1$ | $781 \pm 53$ | $114 \pm 22$ | n.d. | n.d. | n.d. |
| Mandau | $48.1 \pm 0.8$ | $4.8 \pm 0.7$ | $30.3 \pm 2.3$ | $2,484 \pm 669$ | $63 \pm 25$ | n.d. | n.d. | n.d. |
| Tapung Kanan | $53.4 \pm 0.9$ | $5.8 \pm 0.7$ | $30.3 \pm 1.0$ | $1,526 \pm 169$ | $86 \pm 27$ | n.d. | n.d. | n.d. |
| Tapung Kiri | $3.9 \pm 0.1$ | $6.3 \pm 0.5$ | $30.8 \pm 2.2$ | $640 \pm 162$ | $132 \pm 50$ | n.d. | n.d. | n.d. |
| Rajang | $7.7 \pm 0.1$ | $6.7 \pm 0.1$ | $28.8 \pm 1.2$ | $169 \pm 32$ | $190 \pm 26$ | $92 \pm 16$ | $9 \pm 1$ | $1.9 \pm 1.8$ |
| Maludam | $90.7 \pm 1.5$ | $3.8 \pm 0.2$ | $26.0 \pm 0.5$ | $4,031 \pm 805$ | $55 \pm 36$ | $281 \pm 30$ | $5 \pm 2$ | $6.5 \pm 3.2$ |
| Sebuyau | $60.7 \pm 1.0$ | $4.2 \pm 0.2$ | $27.8 \pm 0.6$ | $3,026 \pm 1,047$ | $61 \pm 26$ | $279 \pm 34$ | $9 \pm 5$ | $6.4 \pm 4.9$ |
| Simunjan$_1^*$ | $42.9 \pm 0.7$ | $5.3 \pm 0.4$ | $28.2 \pm 0.6$ | $1,533 \pm 559$ | $107 \pm 21$ | $248 \pm 54$ | $11 \pm 5$ | $5.7 \pm 4.9$ |
| Simunjan$_2^*$ | $42.9 \pm 0.7$ | $5.0 \pm 0.3^{**}$ | $27.9 \pm 0.3$ | $8,366 \pm 1,694$ | $52 \pm 19^{**}$ | $475 \pm 67^{**}$ | $11 \pm 5$ | $11.2 \pm 6.5^{**}$ |

Values are means of river campaigns. Data variability is given by the standard deviation of the measurements. $^*$For the Simunjan, the March 2015 and July 2017 campaigns (Simunjan$_1$) were separated from the January 2016 and March 2017 campaigns (Simunjan$_2$) due to strong differences in the parameters. $^{**}$Due to technical problems during the Simunjan campaign in January 2016, these values are only based on one measurement campaign.

**Table 4.** Data measured in the four Simunjan campaigns.

| | Campaign | $p$H | DOC (mmol L$^{-1}$) | $CO_2$ ($\mu$mol L$^{-1}$) | $O_2$ ($\mu$mol L$^{-1}$) | $CaCO_3$ (mg L$^{-1}$) |
|---|---|---|---|---|---|---|
| Simunjan$_1$ | Mar 2015 | $5.2 \pm 0.3$ | $1.7 \pm 0.7$ | $268 \pm 71$ | $99 \pm 10$ | n.d. |
| Simunjan$_2$ | Jan 2016 | $4.5 \pm 0.3^*$ | $9.4 \pm 1.2$ | $> 330^{**}$ | $139 \pm 9^*$ | $0.52 \pm 0.34$ |
| Simunjan$_2$ | Mar 2017 | $5.0 \pm 0.3$ | $7.4 \pm 0.6$ | $475 \pm 97$ | $52 \pm 19$ | $0.63 \pm 0.64$ |
| Simunjan$_1$ | Jul 2017 | $5.4 \pm 0.3$ | $1.4 \pm 0.3$ | $227 \pm 16$ | $115 \pm 14$ | $0.07 \pm 0.05$ |

Values are means of measurements. Data variability is given by standard deviation of measurements. $^*$Due to technical problems, the March 2017 $p$H, $CO_2$ and $O_2$ data need to be treated cautiously. $^{**}$In March 2017 only a minimum $CO_2$ concentration could be derived.

The Simunjan river shows exceptions to these correlations. Although generally $CO_2$ concentrations stagnate for high peat coverages, extremely high $CO_2$ concentrations were measured during two campaigns in the Simunjan river (Fig. 2). In January 2016 and March 2017 DOC and $CO_2$ concentrations in the Simunjan river were significantly higher than in March 2015 and July 2017 (Simunjan$_1$, Tab. 4). $O_2$ concentrations during these campaigns were lower ($\approx 50\,\mu$mol L$^{-1}$) than for the other Simunjan campaigns ($\approx 107\,\mu$mol L$^{-1}$), while the water $p$H of 5.0 was only slightly lower than during the other campaigns

($p$H $\approx 5.3$). The Simunjan campaigns with high DOC and $CO_2$ concentrations were accompanied by high concentrations of particulate carbonate ($CaCO_3$, Tab. 4), while $CaCO_3$ concentrations in July 2017 were much lower.

## 3.2   Limitation of decomposition rates by $p$H and $O_2$

To gain a better understanding of the $p$H and $O_2$ impacts on decomposition rates, least-squares optimizations of the equations in Tab. 1 (linear $p$H limitation) and Tab. 2 (exponential $p$H limitation) were performed based on measured $p$H, DOC, $CO_2$,
$O_2$ and temperature data. The resulting decomposition parameters for the two $p$H approaches are listed in Tab. 5. A quality assessment of the least-squares optimizations can be found in the appendix D.

For the linear $p$H limitation approach, the decomposition parameters result in a Michaelis constant for $O_2$ limitation of $K_m \approx$ $400\,\mu\text{mol}\,\text{L}^{-1}$, a maximum decomposition rate of $R_{\text{max}} \approx 10\,\mu\text{mol}\,\text{mol}^{-1}\,\text{s}^{-1}$ and a fraction of $O_2$ consumption of $b \approx 90\,\%$ (Tab. 5). Thus, the fraction by which $p$H limits decomposition according to this linear approach ranges from $6\,\%$ in the Batang
Hari to $49\,\%$ in the Maludam, and the limitation by $O_2$ ranges from $71\,\%$ in the Batang Hari to $88\,\%$ in the Maludam and Siak. In total, $O_2$ and $p$H would limit decomposition in a range between 71 and $93\,\%$. Limitation fractions for all rivers are listed in the appendix in Tab. A3.

**Table 5.** Decomposition parameters derived via least-squares optimization.

| parameter | value (lin.) | value (exp.) | unit |
|---|---|---|---|
| $R_{\text{max}}$ | $10 \pm 11$ | $4.0 \pm 0.8$ | $\mu\text{mol}\,\text{mol}^{-1}\,\text{s}^{-1}$ |
| $b$ | $90 \pm 25$ | $81 \pm 10$ | $\%$ |
| $K_m$ | $390 \pm 509$ | $6 \pm 26$ | $\mu\text{mol}\,\text{L}^{-1}$ |
| $\lambda$ | $-$ | $0.52 \pm 0.10$ | |

Data for linear (lin.) and exponential (exp.) approach for the $p$H limitation of decomposition were derived via least-squares optimization of the equations in Tab. 1 and Tab. 2, respectively. $R_{\text{max}}$ is the maximum decomposition rate, $b$ is the fraction of $O_2$ consumption, $K_m$ is the Michaelis constant for $O_2$ limitation and $\lambda$ is the exponential $p$H limitation constant.

For the exponential $p$H limitation approach, the Michaelis constant for $O_2$ ($K_m \approx 6\,\mu\text{mol}\,\text{L}^{-1}$) is significantly smaller than the constant derived for linear $p$H limitation (Tab. 5). The maximum decomposition rate ($R_{\text{max}} \approx 4\,\mu\text{mol}\,\text{mol}^{-1}\,\text{s}^{-1}$) and the
fraction of $O_2$ consumption ($b \approx 81\,\%$) are also smaller than for linear $p$H limitation, but in the same order of magnitude. The exponential $p$H limitation factor results to $\lambda \approx 0.5$. According to these parameters, $O_2$ limits decomposition in the investigated rivers by $\leq 10\,\%$, while the fraction of $p$H limitation ranges from $20\,\%$ in the Batang Hari to $85\,\%$ in the Maludam. The total limitation by $O_2$ and $p$H ranges from 23 to $87\,\%$ (Tab. A4).

To evaluate both decomposition approaches, $CO_2$ and $O_2$ concentrations calculated based on the equations in Tab. 1 and Tab. 2
with the parameters in Tab. 3 and Tab. 5 were compared to measured $CO_2$ and $O_2$ concentrations in the individual rivers. For the linear $p$H limitation approach, correlation coefficients for $CO_2$ and $O_2$ correlations are $R^2 = 0.80$ and $R^2 = 0.88$, respectively

(Fig. 3). For the exponential $p$H limitation approach, the resulting correlation coefficient are similar, whereat the correlation for $CO_2$ ($R^2 = 0.89$) is slightly stronger and the $O_2$ correlation ($R^2 = 0.85$) is slightly weaker (Fig. 4).

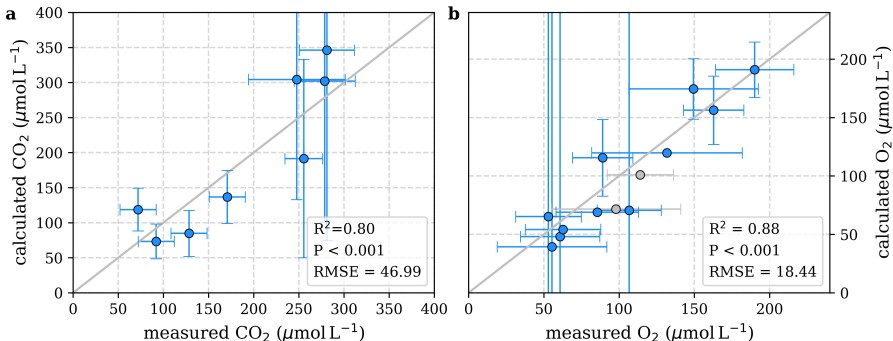

**Figure 3.** Correlation between measured and calculated concentrations of (a) $CO_2$ and (b) $O_2$. Grey lines indicate the 1:1 line. Calculations were performed based on the equations in Tab. 1 which represent linear $p$H limitation of decomposition rates. Each data point represents one river. Grey data points are excluded from the correlation since the data for these rivers are based on less than three campaigns within the same season. This includes the Simunjan campaigns with high carbon concentrations, which are excluded here due to figure scaling and further discussed in the appendix D5.

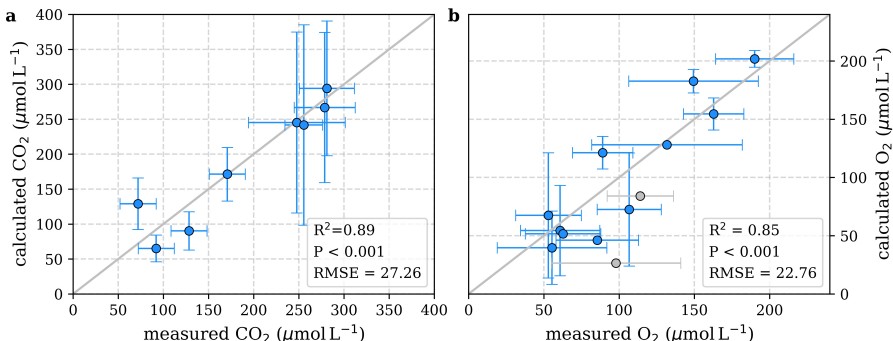

**Figure 4.** Correlation between measured and calculated concentrations of (a) $CO_2$ and (b) $O_2$. Grey lines indicate the 1:1 line. Calculations were performed based on the equations in Tab. 2 which represent exponential $p$H limitation of decomposition rates. Each data point represents one river. Grey data points are excluded from the correlation since the data for these rivers are based on less than three campaigns within the same season. This includes the Simunjan campaigns with high carbon concentrations, which are excluded here due to figure scaling and further discussed in the appendix D5.

## 4 Discussion

### 4.1 Carbon dynamics in peat-draining rivers and their dependencies on peat coverage

The linear correlations observed between peat coverage and DOC (Fig. 2a) as well as between peat coverage and $pH$ (Fig. 2b) agree with results by Wit et al. (2015) and confirm the importance of peat soils as a major DOC source to these rivers, whereas the decomposition of DOC and leaching of organic acids lower the $pH$. The initial increase of $CO_2$ concentrations (Fig. 2c) and decrease of $O_2$ concentrations (Fig. 2d) with peat coverage can be explained by increased DOC decomposition due to higher DOC concentrations and also agrees with the results of Wit et al. (2015).

Previous studies included no data for rivers with peat coverage $> 25\%$ (Wit et al., 2015). In this study, we include additional campaigns at rivers with peat coverages up to $91\%$. We observe that $CO_2$ concentrations in rivers of peat coverage $> 30\%$ level off to fairly constant values for peat coverage $> 50\%$ (Fig. 2c). This agrees with moderate $CO_2$ emissions that were stated for those rivers (Müller et al., 2015; Moore et al., 2013). We find that, according to Eq. (7), the stagnation can be explained by the $pH$ and $O_2$ limitations of decomposition. A similar pattern of stagnating $CO_2$ concentrations has been observed in river sections of high DOC at the Congo river (Borges et al., 2015). The $CO_2$ and DOC concentrations measured in these rivers are comparable to those measured in our study, indicating that the underlying process is valid not only for Southeast Asian rivers but for tropical peat-draining rivers in general.

### 4.2 Decomposition in peat-draining rivers and its dependency on $O_2$ and $pH$

We were able to reproduce the stagnation in $CO_2$ and $O_2$ concentrations by introducing $O_2$ and $pH$ limitations for decomposition rates in the rivers. Model approaches for both linear and exponential $pH$ limitation factors reproduce the observed stagnation in $CO_2$ and $O_2$ concentrations and result in reasonably good correlations with the measured concentrations (Fig. 3 and Fig. 4). However, to evaluate the quality of the two approaches, the resulting parameters need to be further discussed.

The fractions of $O_2$ consumption by decomposition that we derived for both approaches, with $b = (81 \pm 10)\%$ and $b = (90 \pm 25)\%$, agree with the fraction of $0.8$ that was calculated based on the oxygen to carbon ratio in Southeast Asian peat soils (Rixen et al., 2008).

The maximum decomposition rates of $4\,\mu mol\,mol^{-1}\,s^{-1}$ for the exponential approach and $10\,\mu mol\,mol^{-1}\,s^{-1}$ for the linear approach are higher than global soil phenol oxidase activity data published by Sinsabaugh et al. (2008) that stated global average soil phenol oxidase activity of $70.6\,\mu mol\,h^{-1}$ per g organic matter. For a carbon content in organic matter of $38\,mmol\,g^{-1}$ (Sinsabaugh, 2010) this represents approximately $0.5\,\mu mol\,mol^{-1}\,s^{-1}$, while sites of high phenol oxidase activity are listed with up to $3\,\mu mol\,mol^{-1}\,s^{-1}$ (Sinsabaugh et al., 2008). Thus, the derived $R_{max}$ values are slightly higher than measured decomposition rates and therewith in a realistic order of magnitude.

### 4.2.1 Functional dependency of decomposition on $O_2$

The two Michaelis constants for $O_2$ limitation of decomposition, derived for the linear and exponential $pH$ limitation approaches, differ strongly (Tab. 5). As discussed before, the Michaelis constant represents the $O_2$ concentration at which $O_2$ availability limits decomposition by $50\%$. In literature, Michaelis constants between 1 and $40\,\mu mol\,L^{-1}$ are suggested for the $O_2$ impact on phenol oxidase, depending on the phenolic species (Fenoll et al., 2002).

The linear $pH$ limitation approach yields a Michaelis constant of $K_m \approx 390\,\mu mol\,L^{-1}$. This constant is higher than the $O_2$ concentration in atmospheric equilibrium ($\approx 280\,\mu mol\,L^{-1}$), which implies an oxygen deficit at atmospheric conditions that does not exist (Vaquer-Sunyer and Duarte, 2008). However, though the derived $K_m$ value for this linear $pH$ limitation is unrealistically high, this does not necessarily negate the linear $pH$ approach. High parameter interdependence between $K_m$ and $R_{max}$ complicate the computation of these decomposition parameters (appendix D1). To disentangle the impact of the intercorrelated parameters, additional least-squares optimizations at fixed $K_m$ values ranging from 1 to $40\,\mu mol\,L^{-1}$ (Fenoll et al., 2002) were performed (appendix D2). These optimizations result in maximum decomposition rates of $R_{max} = (1.4 - 2.4)\,\mu mol\,mol^{-1}\,s^{-1}$ and $O_2$ consumption factors of $b = (102 - 109)\,\%$ and therewith do not agree with literature values of these parameters ($R_{max} \geqslant 3\,\mu mol\,mol^{-1}\,s^{-1}$ & $b \approx 80\%$; Sinsabaugh et al., 2008; Rixen et al., 2008).

The exponential $pH$ limitation approach yields a Michaelis constant of $K_m \approx 6\,\mu mol\,L^{-1}$. This value is in good agreement with the literature data of 1 to $40\,\mu mol\,L^{-1}$ (Fenoll et al., 2002). Its large uncertainty ($> 400\%$, Tab. 5) is mainly caused by relatively high concentrations of $O_2$ in the rivers. Due to exchange with atmospheric $O_2$ the concentrations in all rivers exceed the median $O_2$ threshold to lethal hypoxic conditions of $50\,\mu mol\,L^{-1}$ (Vaquer-Sunyer and Duarte, 2008). Thus, the $O_2$ limitation in peat-draining rivers is relatively small (between 3 and $10\%$, Tab. A4). Consequentially a majority of the decomposition limitation is caused by the low $pH$ in peat-draining rivers that we found to limit the decomposition rates in rivers of high peat coverage (low $pH$) by up to $85\%$ (Tab. A4).

### 4.2.2 Functional dependency of decomposition on $pH$

Our results indicate the exponential $pH$ limitation of decomposition to be more realistic than the linear $pH$ limitation. The exponential limitation better represents river $CO_2$ especially for high $CO_2$ concentrations which are most strongly effected by the $pH$ limitation. The exponential limitation is additionally supported by the unrealistically high $O_2$ limitation resulting from the linear $pH$ approach. The strong collinearity between decomposition parameters in the linear $pH$ limitation approach complicates the interpretation of the parameters mentioned above. Additional calculations of the parameters $R_{max}$ and $b$ for fixed $K_m$ disagree with literature data and thus further disprove the linear approach (appendix D2).

The exponential $pH$ coefficient results to $\lambda = 0.5 \pm 0.1$. Thus, in terms of $H^+$ activity the correlation is given by $\left\{ H^+ \right\}^{\frac{0.5}{\ln(10)}}$, which roughly equals the fifth root of $H^+$ activity. The derived limitation coefficient is similar to coefficients reported for high latitude peat soils ($\lambda = 0.65$ & $\lambda = 0.77$) that were determined via laboratory measurements of phenol oxidase activity (Williams et al., 2000). The fact that the exponential inhibition by $pH$ can be found in those high latitude peat soils, as well as

in tropical peat-draining rivers suggests that the investigated correlations and processes are also relevant in other regions and that soil and water $p$H are important regulators of global carbon emissions.

### 4.3 Impact of additional processes

Our results neglect the direct leaching of $CO_2$ from soils as well as the photo-mineralization of DOC and the consumption of $CO_2$ by autotrophic production within rivers. $CO_2$ leaching rates are likely higher for peat soils than for mineral soils (Kang et al., 2018; Abril and Borges, 2019) and autotropic production is limited in peat-draining rivers (Wit et al., 2015). Thus, both of these processes would work against the observed stagnation in $CO_2$ concentrations, and the exclusion of these processes could cause underestimation of the limitation factors rather than overestimation.

A recent study by Nichols and Martin (2021) found low phenol oxidase activity in Southeast Asian peat-draining rivers amd low degradation of DOC from those rivers in an additional incubation experiment. They concluded that that the remineralization of peat-derived DOC in Southeast Asian aquatic systems is likely dependent on photodegradation rather than microbial respiration (Nichols and Martin, 2021). This is supported by photolability of DOC from those regions (Martin et al., 2018). However, photomineralization rates would not be impacted by river $p$H or $O_2$. Thus, with photomineralization as the main cause of DOC degradation, no stagnation in $CO_2$ is expected. Accordingly, photomineralization of DOC, like the before-mentioned processes, would work against the observed $CO_2$ stagnation and could cause underestimation of the limitation parameters.

### 4.4 Disruption of the $p$H limitation by carbonates

Typically, concentrations of particulate carbonate in peat-draining rivers are low (Wit et al., 2018). However we observed high $CaCO_3$ concentrations for two of the four campaigns. These two campaigns also show high DOC and $CO_2$ concentrations (Tab. 4). Possible causes for high carbonate concentrations during these campaigns could be increased erosion of mineral soils due to deforestation in mountain regions upstream or liming practices in plantations along the river. In either case, high carbonate concentrations at such a low $p$H indicate high dissolution of carbonates which might have counteracted a more prominent decrease in $p$H due to decomposition of DOC. At the same time, the low river $p$H causes transformation of dissolved carbonates to $CO_2$ and thus additionally increases $CO_2$ concentrations. These processes seem to have suspended the natural $p$H limitation of $CO_2$ production in peat-draining rivers and explain the high $CO_2$ concentrations observed during those two Simunjan campaigns (Tab. 4). Calculation based on the derived decomposition dependencies would indicate even higher $CO_2$ concentrations than measured. This indicates that the river carbon parameters were not in thermodynamic equilibrium during these campaigns as is further discussed in appendix D5.

### 5 Conclusions

Our study shows that $CO_2$ concentrations in and emissions from Southeast Asian rivers stagnate for high peat coverages of the river catchments. Despite further increase in river DOC concentrations, $CO_2$ concentrations are fairly constant for peat

coverages $> 50\%$. We find that this stagnation is caused by a natural limitation of DOC decomposition in these rivers. This process provides an explanation to moderate $CO_2$ emissions measured from rivers of high carbon content

Correlation to measured data indicates that the limitation in decomposition is mainly caused by low river $pH$. Data reveal an exponential limitation of DOC decomposition by $pH$ as the most realistic scenario. This reduces the $CO_2$ production in rivers of high peat coverage by up to $85\%$. The limiting impact of $O_2$ on decomposition in the rivers results to be comparatively small with $< 10\%$. This derived limitation of decomposition should be included to improve model studies and accurately capture river $CO_2$ emissions from tropical peat areas.

Campaigns with high carbon loads in the Simunjan River indicate that the natural $CO_2$ limitation can be suspended by high input of DOC and carbonates. Data from campaigns with enhanced concentrations of DOC and suspended carbonates reveal $CO_2$ emissions that were increased by almost $100\%$. Here, the high DOC concentrations enhance decomposition and the input of carbonates counteract the $pH$ decrease associated with large inputs of $CO_2$. Possible sources for enhanced carbonate concentrations can be soil erosion upstream of coastal peatlands, or liming practices in plantations along the rivers, which are common practice to improve plant growth on acidic soils. This carbonate impact should be considered for anthropogenic activities like liming and enhanced weathering.

Our study is based on measurements in Southeast Asian peat-draining rivers. However, comparison to data from African rivers and laboratory studies of decomposition in temperate peat soils suggest that the investigated correlations and processes are also relevant in other regions and that soil and water $pH$ are important regulators of global carbon emissions.

*Code and data availability.* Averaged data from the river campaigns investigated in this study as well as the python code for the performed least-squares approximations are available as a supplementary files to this paper. Raw data from the measurement campaigns are available at the Institute of Environmental Physics, University of Bremen, Bremen, Germany, and will be provided upon request

*Author contributions.* AK performed the analysis and led the writing of the paper jointly with TR and TW. DM provided calculations of catchment parameters and in-depth comments on the manuscript. MM coordinated the field data collection in Malaysia. JN contributed to the data interpretation. All authors discussed results and commented on the manuscript.

*Competing interests.* The authors declare that they have no conflict of interest

*Acknowledgements.* We are grateful to the Sarawak Forestry Department and Sarawak Biodiversity Centre for permission to conduct collaborative research in Sarawak under permit numbers NPW.907.4.4(Jld.14)-161, SBC-RA-0097-MM, and Park Permit WL83/2017.

## Appendix A:  Additional Figures & Tables

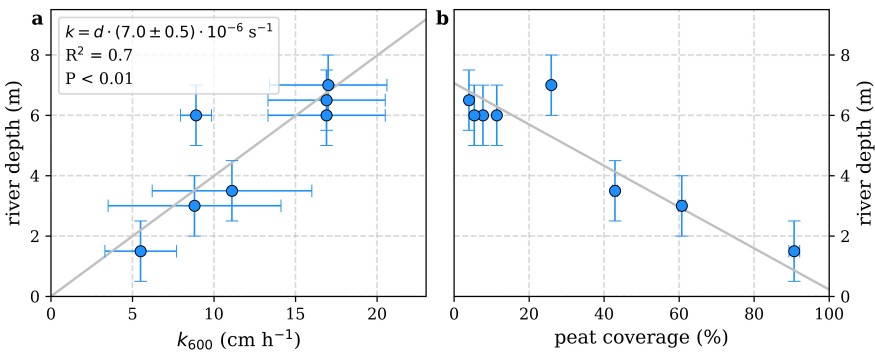

**Figure A1.** Correlation of river depth with (a) atmospheric exchange coefficients ($k_{600}$) and (b) catchment peat coverage. A linear correlation between river depth and exchange coefficient reveals a slope of $k_{600}/d = (2.5 \pm 0.2)\,\mathrm{cm\,h^{-1}\,m^{-1}} = (7.0 \pm 0.5) \cdot 10^{-6}\,\mathrm{s^{-1}}$.

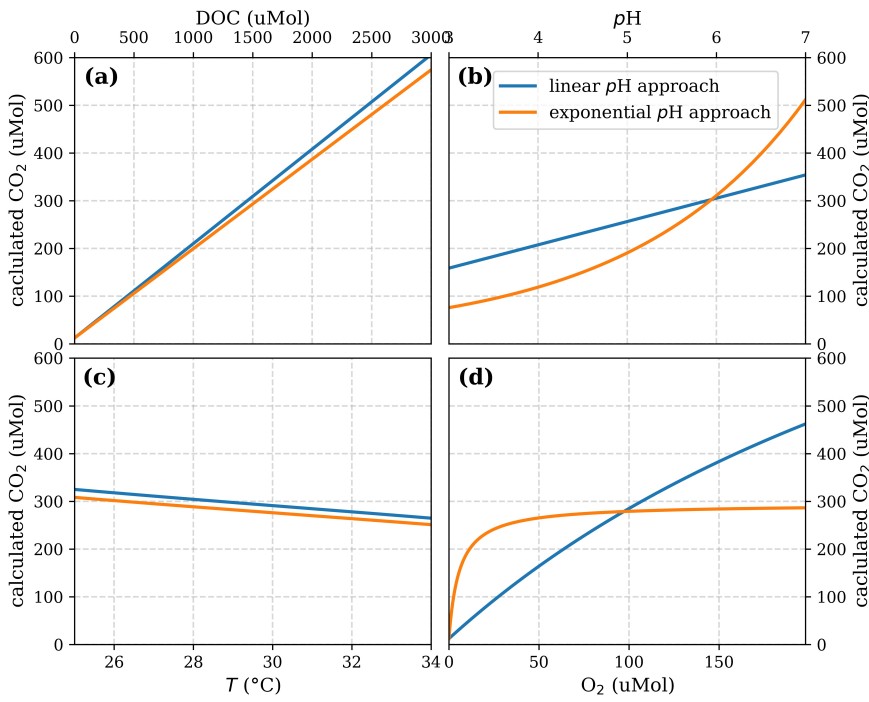

**Figure A2.** Functional dependencies of $CO_2$ on (a) DOC, (b) $p$H, (c) temperature ($T$) and (d) $O_2$ according to the equation in Tab. 1 (linear $p$H approach) and in Tab. 2 (exponential $p$H approach).

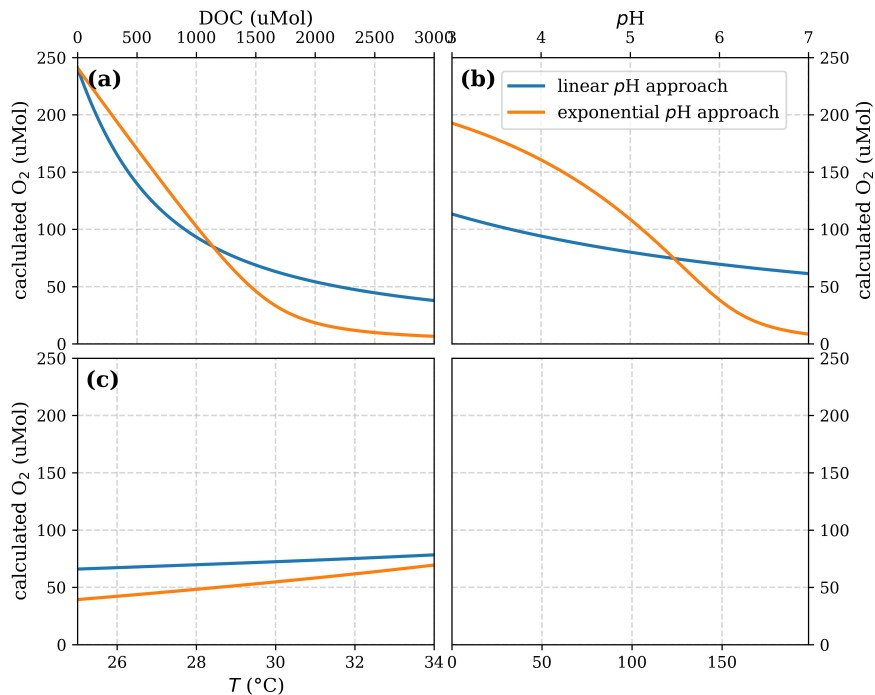

**Figure A3.** Functional dependencies of $O_2$ on (a) DOC, (b) $p$H and (c) temperature ($T$) according to the equation in Tab. 1 (linear $p$H approach) and in Tab. 2 (exponential $p$H approach).

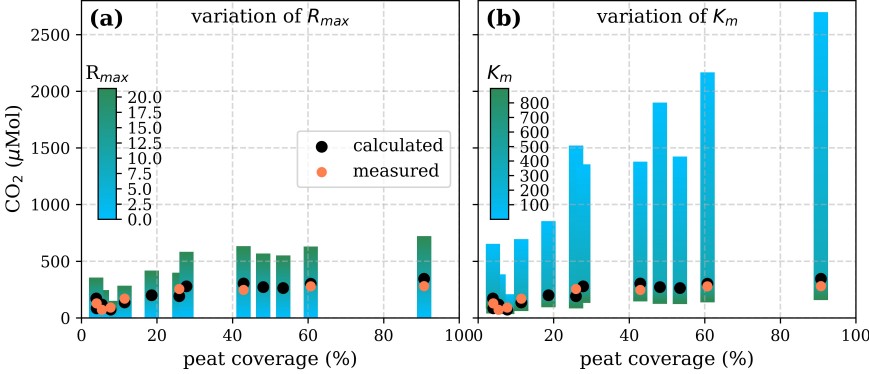

**Figure A4.** Sensitivity of calculated $CO_2$ on (a) the maximum decomposition rate ($R_{max}$) and (b) the Michaelis constant for $O_2$ concentration ($K_m$) according to the equation in Tab 1 for the linear $p$H approach and the parameter ranges given in Tab. 5.

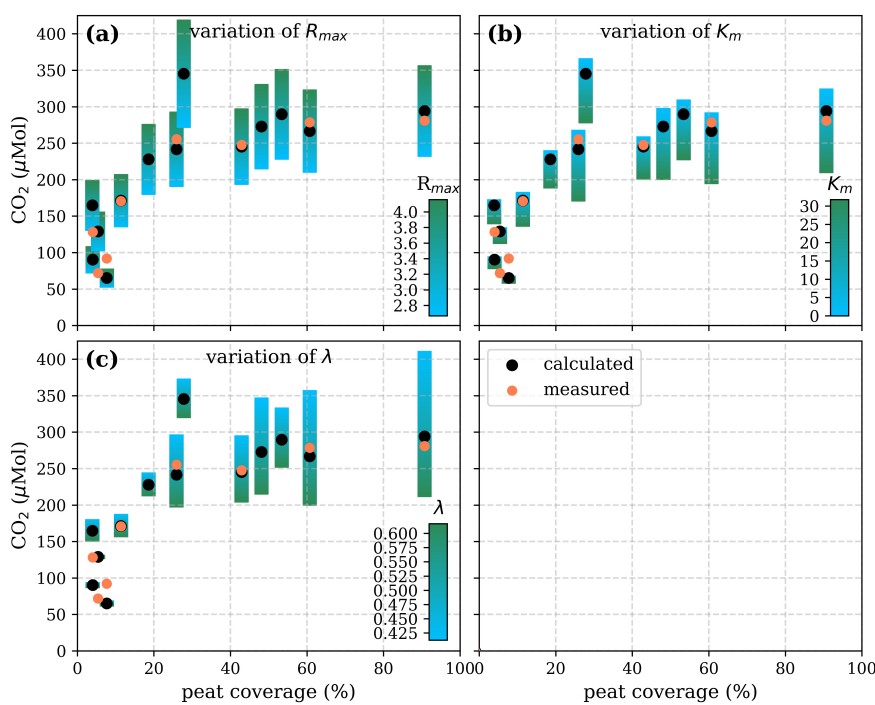

**Figure A5.** Sensitivity of calculated $CO_2$ on (a) the maximum decomposition rate ($R_{\max}$), (b) the Michaelis constant for $O_2$ concentration ($K_m$) and (c) the exponential $p$H limitation constant ($\lambda$) according to the equation in Tab 2 for the exponential $p$H approach and the parameter ranges given in Tab. 5.

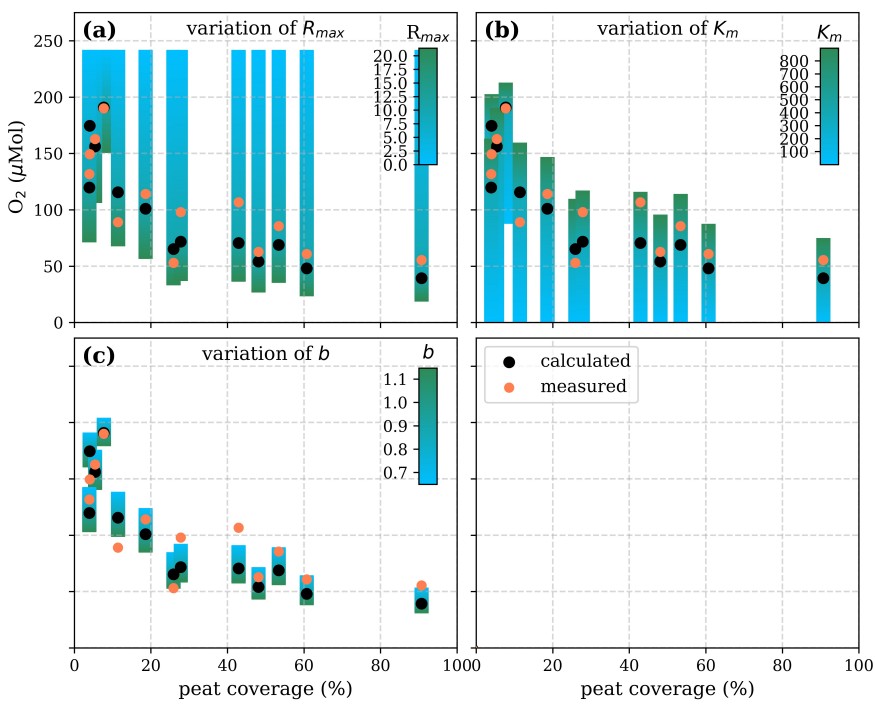

**Figure A6.** Sensitivity of calculated $O_2$ on (a) the maximum decomposition rate ($R_{max}$), (b) the Michaelis constant for $O_2$ concentration ($K_m$) and (c) the fraction of $O_2$ consumption by decmposition ($b$) according to the equation in Tab 1 for the linear $p$H approach and the parameter ranges given in Tab. 5.

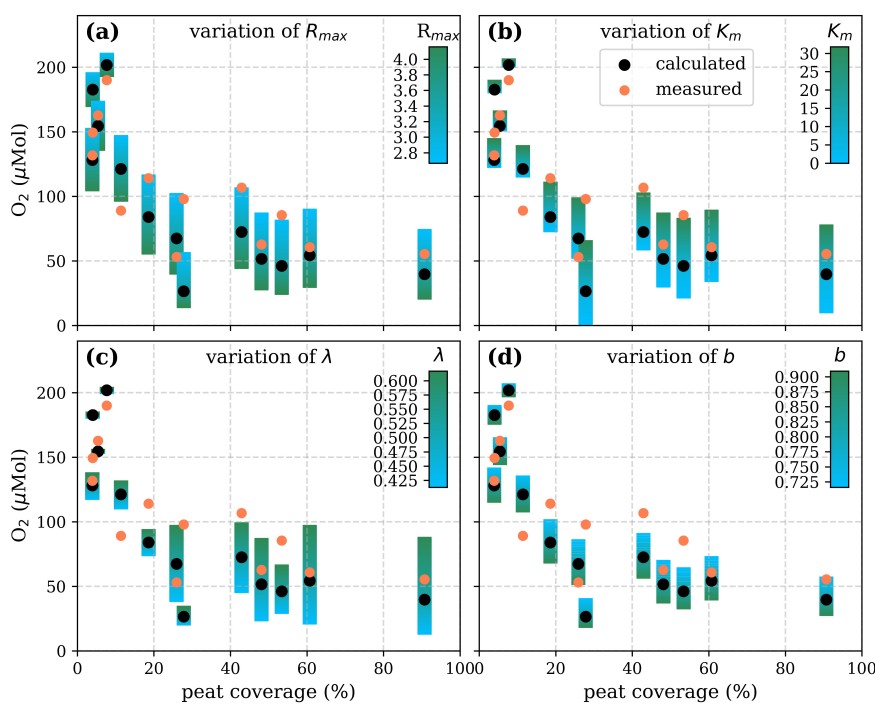

**Figure A7.** Sensitivity of calculated $O_2$ on (a) the maximum decomposition rate ($R_{max}$), (b) the Michaelis constant for $O_2$ concentration ($K_m$), (c) the exponential $p$H limitation constant ($\lambda$) and (d) the fraction of $O_2$ consumption by decomposition according to the equation in Tab 2 for the exponential $p$H approach and the parameter ranges given in Tab. 5.

Table A1. List of river campaigns

| River | 03.04 | 09.04 | 08.05 | 03.06 | 04.06 | 11.06 | 03.08 | 10.09 | 10.12 | 04.13 | 03.14 | 03.15 | 01.16 | 08.16 | 03.17 | 07.17 |
|---|---|---|---|---|---|---|---|---|---|---|---|---|---|---|---|---|
| Maludam | – | – | – | – | – | – | – | – | – | – | ✓ | ✓ | ✓ | – | ✓ | ✓ |
| Sebuyau | – | – | – | – | – | – | – | – | – | – | – | ✓ | ✓ | – | ✓ | ✓ |
| Simunjan | – | – | – | – | – | – | – | – | – | – | – | ✓ | ✓ | – | ✓ | ✓ |
| Rajang | – | – | – | – | – | – | – | – | – | – | – | – | ✓ | ✓ | ✓ | – |
| Musi | – | – | – | – | – | – | – | ✓ | ✓ | ✓ | – | – | – | – | – | – |
| Batang Hari | – | – | – | – | – | – | – | ✓ | ✓ | ✓ | – | – | – | – | – | – |
| Indragiri | – | – | – | – | – | – | – | ✓ | – | ✓ | – | – | – | – | – | – |
| Kampar | – | – | – | – | ✓ | – | ✓ | – | – | – | – | – | – | – | – | – |
| Rokan | – | – | – | – | ✓ | – | ✓ | – | – | – | – | – | – | – | – | – |
| Siak | ✓ | ✓ | ✓ | ✓ | – | ✓ | – | ✓ | – | ✓ | – | – | – | – | – | – |
| Mandau | ✓ | ✓ | ✓ | ✓ | – | – | – | – | – | – | – | – | – | – | – | – |
| Tapung Kanan | ✓ | ✓ | ✓ | ✓ | – | – | – | – | – | – | – | – | – | – | – | – |
| Tapung Kiri | ✓ | ✓ | ✓ | ✓ | – | – | – | – | – | – | – | – | – | – | – | – |

**Table A2.** Temperature dependent exchange coefficients $k$ and Henry coefficients $K$ of $CO_2$ and $O_2$ for the individual rivers.

| River | $T$ (°C) | $k_{CO_2}$ (cm h$^{-1}$) | $k_{O_2}$ (cm h$^{-1}$) | $K_{CO_2}$ (mmol L$^{-1}$ atm$^{-1}$) | $K_{O_2}$ (mmol L$^{-1}$ atm$^{-1}$) |
|---|---|---|---|---|---|
| Musi | $30.6 \pm 0.3$ | $23 \pm 6$ | $26 \pm 7$ | $29.4 \pm 0.2$ | $1.16 \pm 0.10$ |
| Batang Hari | $30.0 \pm 0.1$ | $22 \pm 5$ | $25 \pm 6$ | $29.8 \pm 0.1$ | $1.17 \pm 0.02$ |
| Indragiri | $31.5 \pm 0.1$ | $23 \pm 6$ | $26 \pm 6$ | $28.8 \pm 0.1$ | $1.15 \pm 0.03$ |
| Siak | $30.0 \pm 0.2$ | $22 \pm 6$ | $25 \pm 7$ | $29.9 \pm 0.2$ | $1.18 \pm 0.08$ |
| Kampar | $29.4 \pm 0.7$ | n.d. | n.d. | $30.3 \pm 0.5$ | $1.19 \pm 0.28$ |
| Rokan | $28.9 \pm 1.1$ | n.d. | n.d. | $30.7 \pm 0.8$ | $1.20 \pm 0.43$ |
| Mandau | $30.3 \pm 2.3$ | n.d. | n.d. | $29.6 \pm 1.7$ | $1.17 \pm 0.89$ |
| Tapung Kanan | $30.3 \pm 1.0$ | n.d. | n.d. | $29.6 \pm 0.7$ | $1.17 \pm 0.37$ |
| Tapung Kiri | $30.8 \pm 2.2$ | n.d. | n.d. | $29.2 \pm 1.6$ | $1.16 \pm 0.82$ |
| Rajang | $28.8 \pm 1.2$ | $11 \pm 5$ | $13 \pm 5$ | $30.8 \pm 0.9$ | $1.20 \pm 0.47$ |
| Maludam | $26.0 \pm 0.5$ | $6 \pm 3$ | $6 \pm 3$ | $33.1 \pm 0.5$ | $1.25 \pm 0.23$ |
| Sebuyau | $27.8 \pm 0.6$ | $11 \pm 7$ | $12 \pm 7$ | $31.5 \pm 0.5$ | $1.21 \pm 0.24$ |
| Simunjan$_1^*$ | $28.2 \pm 0.6$ | $13 \pm 7$ | $14 \pm 7$ | $31.2 \pm 0.9$ | $1.21 \pm 0.25$ |
| Simunjan$_2^*$ | $27.9 \pm 0.3$ | $13 \pm 6$ | $14 \pm 7$ | $31.5 \pm 0.2$ | $1.21 \pm 0.13$ |

Exchange coefficients were derived from measured $k_{600}$ (Tab. 3) and water temperature ($T$) according to $k_X = k_{600} \cdot (Sc_X/600)^{-n}$ with Schmidt numbers $Sc_{CO_2}$ and $Sc_{O_2}$ derived according to the equations in (Wanninkhof, 1992). An exponent of $n = 1/2$ (valid for rough surfaces, Zappa et al., 2007) was used for the rivers. Henry coefficients were derived based on water temperature from the equations stated in (Weiss, 1974) for $CO_2$ and the equations stated in (Weiss, 1970) for $O_2$.

**Table A3.** $pH$ and $O_2$ limitations in the individual rivers based on linear $pH$ approach.

| River | $pH$ lim. (%) | $O_2$ lim. (%) | total lim. (%) | River | $pH$ lim. (%) | $O_2$ lim. (%) | total lim. (%) |
|---|---|---|---|---|---|---|---|
| Musi | $9 \pm 3$ | $87 \pm 23$ | $93 \pm 2$ | Tapung Kanan | $23 \pm 9$ | $82 \pm 26$ | $91 \pm 10$ |
| Batang Hari | $6 \pm 4$ | $71 \pm 32$ | $72 \pm 32$ | Tapung Kiri | $16 \pm 6$ | $75 \pm 37$ | $79 \pm 19$ |
| Indragiri | $16 \pm 4$ | $81 \pm 25$ | $84 \pm 9$ | Rajang | $11 \pm 2$ | $67 \pm 34$ | $71 \pm 20$ |
| Siak | $32 \pm 7$ | $88 \pm 19$ | $92 \pm 4$ | Maludam | $49 \pm 2$ | $87 \pm 23$ | $93 \pm 2$ |
| Kampar | $14 \pm 5$ | $80 \pm 32$ | $83 \pm 13$ | Sebuyau | $44 \pm 3$ | $87 \pm 22$ | $92 \pm 3$ |
| Rokan | $13 \pm 2$ | $77 \pm 28$ | $83 \pm 9$ | Simunjan | $29 \pm 2$ | $79 \pm 27$ | $85 \pm 8$ |
| Mandau | $36 \pm 10$ | $86 \pm 22$ | $91 \pm 10$ | | | | |

Fraction by which the decomposition is lowered due to the impact of $pH$ and $O_2$, calculated based on the limitation factors in Eq. (7) and the parameters in Tab. 5 according to $pH$ lim. $= (1 - L_{pH})$, $O_2$ lim. $= (1 - L_{O_2})$ and total lim. $= (1 - L_{pH} \cdot L_{O_2})$.

**Table A4.** $pH$ and $O_2$ limitations in the individual rivers based on exponential $pH$ approach.

| River | $pH$ lim. (%) | $O_2$ lim. (%) | total lim. (%) | River | $pH$ lim. (%) | $O_2$ lim. (%) | total lim. (%) |
|---|---|---|---|---|---|---|---|
| Musi | $28 \pm 5$ | $4 \pm 1$ | $31 \pm 5$ | Tapung Kanan | $59 \pm 7$ | $7 \pm 2$ | $62 \pm 7$ |
| Batang Hari | $20 \pm 3$ | $4 \pm 1$ | $23 \pm 4$ | Tapung Kiri | $46 \pm 6$ | $4 \pm 1$ | $49 \pm 7$ |
| Indragiri | $46 \pm 6$ | $6 \pm 2$ | $50 \pm 7$ | Rajang | $34 \pm 5$ | $3 \pm 1$ | $36 \pm 5$ |
| Siak | $71 \pm 7$ | $10 \pm 5$ | $74 \pm 8$ | Maludam | $85 \pm 5$ | $10 \pm 4$ | $87 \pm 6$ |
| Kampar | $43 \pm 6$ | $6 \pm 1$ | $46 \pm 7$ | Sebuyau | $83 \pm 6$ | $9 \pm 4$ | $83 \pm 6$ |
| Rokan | $40 \pm 6$ | $5 \pm 1$ | $43 \pm 6$ | Simunjan | $68 \pm 7$ | $5 \pm 1$ | $70 \pm 7$ |
| Mandau | $76 \pm 7$ | $9 \pm 3$ | $78 \pm 7$ | | | | |

Fraction by which the decomposition is lowered due to the impact of $pH$ and $O_2$, calculated based on the limitation factors in Eq. (7) and the parameters in Tab. 5 according to $pH$ lim. $= (1 - L_{pH})$, $O_2$ lim. $= (1 - L_{O_2})$ and total lim. $= (1 - L_{pH} \cdot L_{O_2})$.

## Appendix B:  Impact of data limitation on study results

### B1  Impact of sampling location

The data for this study was collected from samples taken in river sections that flow through peat soil. This ensures that the impact of peat soils on the river parameters is captured.

Concentrations measured in the small Malaysian rivers (Maludam and Sebuyau and Simunjan), with the exception of the Simunjan campaigns in Jamuary 2016 and March 2017 (Tab. 4, Fig. B3), show little variation over the river path and between campaigns (Fig. B1, B2 B3). However, the larger rivers drain mineral soils for the majority of their path and only reach peat regions close to the coast. Those rivers exhibit stronger differences in carbon concentrations along the length of the river. Rixen et al. (2010) found that DOC concentrations in the Siak river are by a factor of up to 4 higher in coastal peat regions than in the upstream river. $CO_2$ concentrations in the large Sumatran rivers were not measured outside of the coastal peat regions. Due to the lower $p$H in river parts that cut through peat and the related $p$H limitation of DOC decomposition, the difference in $CO_2$ concentrations along the river is likely lower than the difference in DOC concentrations. This is also indicated by $CO_2$ measurements in the Rajang river that reveal $CO_2$ concentrations in the peat-draining rivers sections to be only $(15-20)\%$ higher than $CO_2$ concentrations upstream the peat regions (Müller-Dum et al., 2019).

### B2  Impact of seasonality

The Southeast Asian study area is impacted by the Malaysian-Australian monsoon that causes presence of moisture loaded air with high precipitation rates from October to April while dry air dominates from May to September. To catch the impact of these rain and dry seasons on river carbon dynamics, campaigns in different months of the year were performed (Tab. A1). However, the seasonal data coverage is not dense enough to clearly identify or disprove a seasonal pattern in the measured data (Fig. B4, B5 & B6).

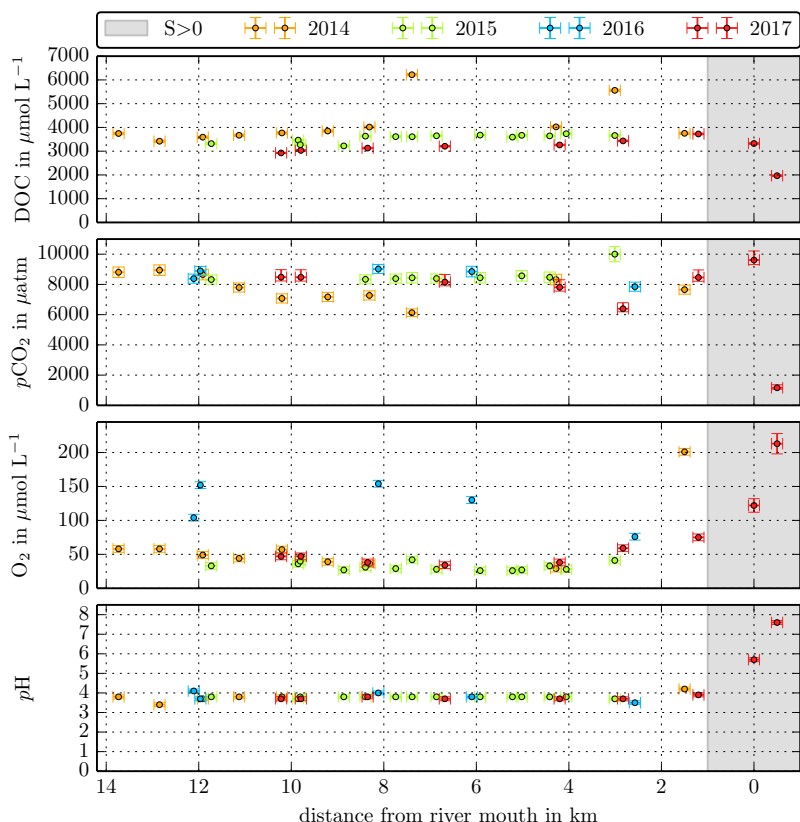

**Figure B1.** Individual DOC, CO2, O2 and $p$H measurements in the Maludam river versus the distance from the river mouth. Different colors stand for the individual river campaigns and the gray shaded area indicates regions of $S > 1$ that were excluded from the data in this study.

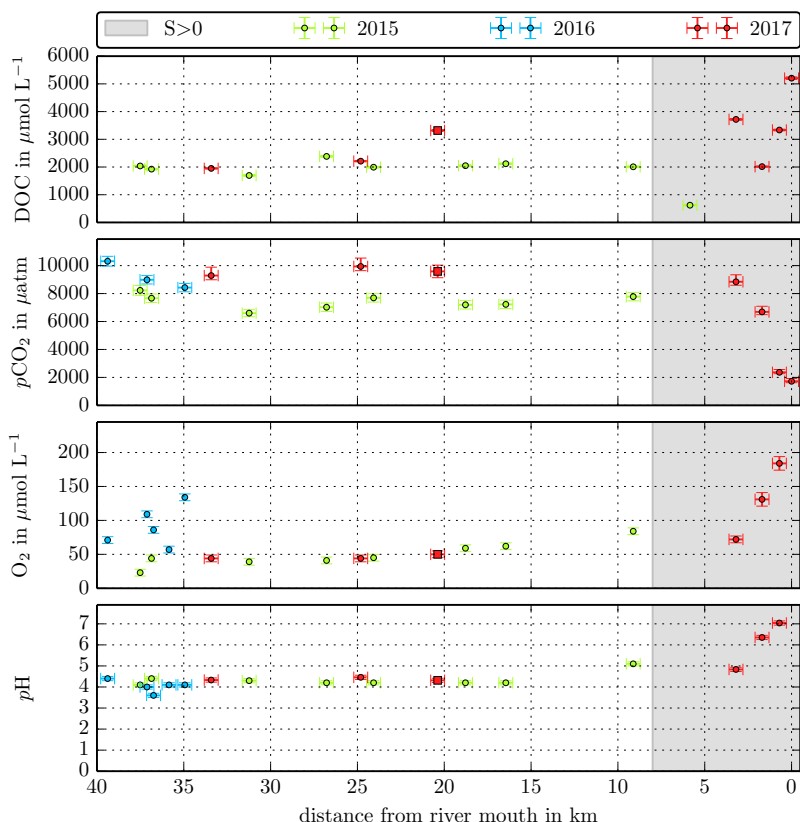

**Figure B2.** Individual DOC, CO2, O2 and $p$H measurements in the Sebucau river versus the distance from the river mouth. Different colors stand for the individual river campaigns and the gray shaded area indicates regions of $S > 1$ that were excluded from the data in this study.

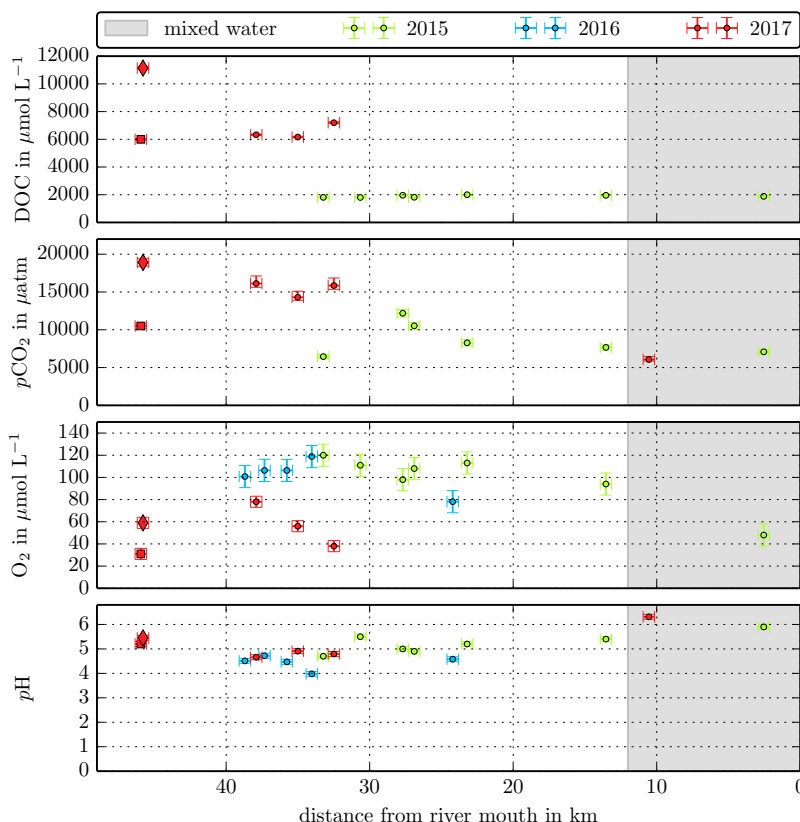

**Figure B3.** Individual DOC, CO2, O2 and $p$H measurements in the Simunjan river versus the distance from the river mouth. Different colors stand for the individual river campaigns and the gray shaded area indicates regions of $S > 1$ that were excluded from the data in this study.

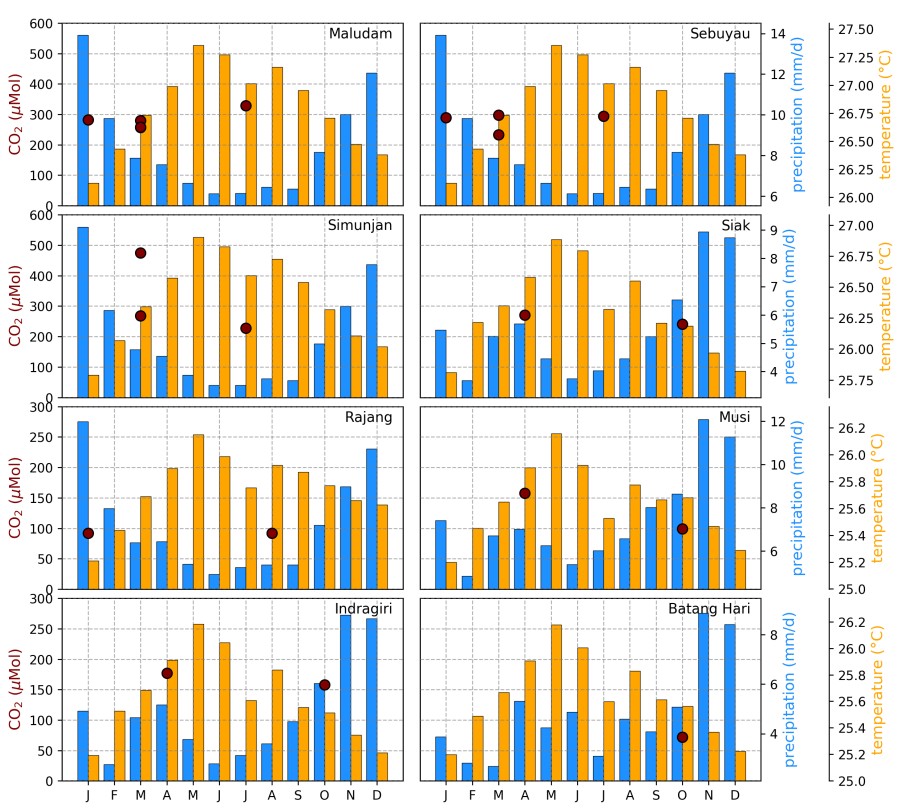

**Figure B4.** Average $CO_2$ concentrations for individual campaigns compared to monthly temperature and precipitation data (2005-2015 average) at the location of the respective river. Each panel represents one river.

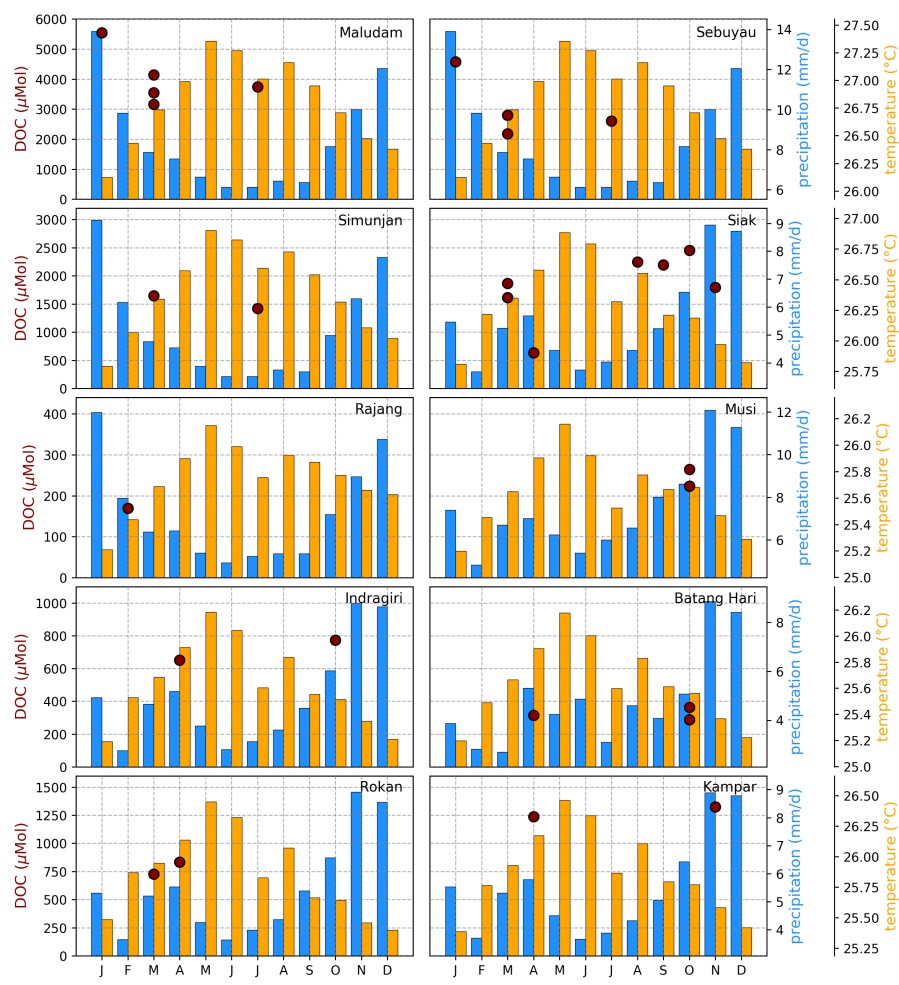

**Figure B5.** Average DOC concentrations for individual campaigns compared to monthly temperature and precipitation data (2005-2015 average) at the location of the respective river. Each panel represents one river.

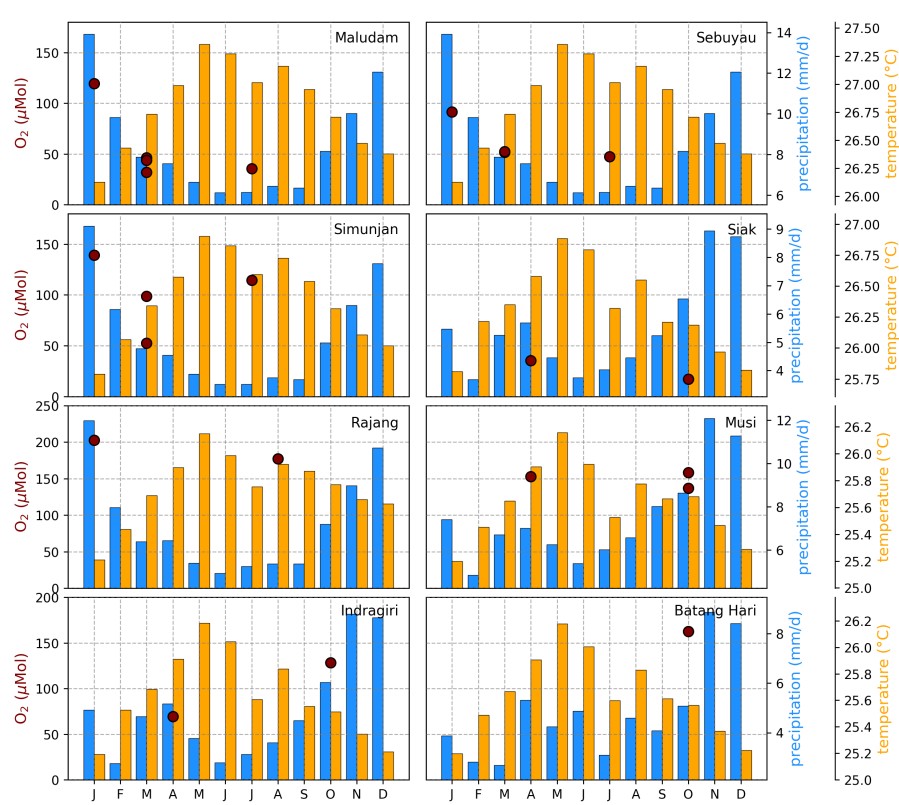

**Figure B6.** Average O₂ concentrations for individual campaigns compared to monthly temperature and precipitation data (2005-2015 average) at the location of the respective river. Each panel represents one river.

## Appendix C: Comparison of different peat coverage estimates

Different peat maps are available for Southeast Asia and the approaches to determine peat coverage of river catchments were inconsistent among different studies cited in our paper. We want to show here that the choice of a data product is crucial for the determination of peat coverage. We are comparing three different products (Tab. C1): The FAO Digital Soil Map of the World, Country products downloaded at Global Forest Watch and the Center for International Forestry Research (CIFOR) Wetlands distribution.

**Table C1.** Different data products used to assess peatland extent in the catchments.

| | **FAO** |
|---|---|
| **Product** | Food and Agriculture Organization of the United Nations (FAO): Digital Soil Map of the World |
| **Coordinate System** | WGS 1984 |
| **Reference** | FAO Land and Water Development Division. Digital Soil Map of the World. Version 3.6. Rome, Italy 2003. |
| **Website** | http:www.fao.org/geonetwork/srv/en/metadata.show?id=14116 |
| **Notes** | Peatlands were identified as Histosols. On Sumatra and Borneo, these are Dystric Histosols ("Od") |
| | **GFW** |
| **Product** | Global Forest Watch Country products |
| **Coordinate System** | WGS 1984 |
| **Reference** | Indonesia: Ministry of Agriculture. Indonesia peat lands, 2012. |
| | Malaysia: Wetlands International. "Malaysia peat lands", 2004. |
| **Website** | www.globalforestwatch.org |
| | **CIFOR** |
| **Product** | Center for International Forestry Research (CIFOR): Tropical and Subtropical Wetlands Distribution version 2. |
| **Coordinate System** | WGS 1984 |
| **Reference** | Data product: Gumbricht et al. (2018); Related publication: Gumbricht et al. (2017) |
| **Website** | https://data.cifor.org/dataset.xhtml?persistentId=doi:10.17528/CIFOR/DATA.00058 |
| **Notes** | Of the three available files, the product used was TROP_SUBTROP_PeatV21_2016_CIFOR.7z |

Those three products lead to highly different results (Tab. C2). We observed a tendency that CIFOR leads to smaller peat coverage than FAO and GFW. This is because CIFOR misses some, but not all peat areas that are known to be under industrial plantations. Gumbricht et al. (2017) already pointed out that their model underestimates peatland area in Sumatra because peats are largely drained, which the model does not capture. However, in the Musi and Batang Hari catchment, CIFOR sees larger peat areas than FAO and GFW, which means that some peatlands might be missing in those maps.

**Table C2.** Results for peat coverage (PC) in the different catchments using the three different data products.

| River name | Catchment ($km^2$) | PC GFW | PC CIFOR | PC FAO |
|---|---|---|---|---|
| Batang Hari | 43,778 | $5.4 \pm 0.1$ | $6.8 \pm 0.1$ | $5.0 \pm 0.1$ |
| Indragiri | 17,713 | $11.4 \pm 0.2$ | $9.6 \pm 0.1$ | $8.6 \pm 0.1$ |
| Kampar | 23,610 | $27.8 \pm 0.4$ | $20.2 \pm 0.2$ | $18.8 \pm 0.3$ |
| Musi | 57,602 | $4.0 \pm 0.1$ | $11.3 \pm 0.1$ | $3.7 \pm 0.1$ |
| Rokan | 19,953 | $18.4 \pm 0.3$ | $8.8 \pm 0.1$ | $30.3 \pm 0.5$ |
| Siak | 11,719 | $25.9 \pm 0.4$ | $14.8 \pm 0.1$ | $27.2 \pm 0.4$ |
| Maludam | 91 | $90.7 \pm 1.4$ | $82.3 \pm 1.1$ | $100.0 \pm 1.5$ |
| Rajang | 51,699 | $7.7 \pm 0.1$ | $7.4 \pm 0.1$ | $10.6 \pm 0.2$ |
| Sebuyau | 451 | $60.7 \pm 0.9$ | $41.2 \pm 0.4$ | $75.8 \pm 1.2$ |
| Simunjan | 755 | $42.9 \pm 0.7$ | $20.3 \pm 0.2$ | $25.9 \pm 0.4$ |

We decided to use the GFW maps for several reasons: 1) CIFOR seems to miss peat under industrial plantations, which is still relevant for river carbon dynamics. Therefore, we chose not to use the CIFOR maps. 2) Between GFW and FAO, GFW is more

recent than FAO for Indonesia. For Sarawak (Malaysia), both are based on the 1968 soil map by the Land Survey Department, but FAO uses a 10-fold coarser scale than the 1968 soil map (1:5,000,000 compared to 1:500,000). Thus, the GFW product was used. & 3) GFW maps are based on official information, and we believe that the local authorities would know best about the peatland distribution in their country.

Similar to the peat coverage, the publications from which we use data in our study all had different approaches to determining

catchment size – either including (Müller-Dum et al., 2019) or excluding (Wit et al., 2015) smaller sub-catchments. In our study, we aimed to unify those different approaches. Therefore, we recalculated catchment areas from one single data product (HydroSHEDS, (Lehner et al., 2006)) including sub-catchments that were identified using HydroSHEDS flow directions. The Simunjan catchment is included in the bigger Sadong catchment in HydroSHEDS. Therefore, it was manually delineated using HydroSHEDS flow directions.

## Appendix D: Quality assessment of least-squares optimizations

Uncertainty sources in the least-squares optimizations are interdependencies between the fitted parameters and noise in the measured data. We try to minimize the impact of measurement noise by including relative uncertainties ($\sigma$) of measured $CO_2$ and $O_2$ in the least-squares optimization. Thus, data from rivers with higher variation in measured parameters are constrained less rigidly in the optimization. The parameter interdependence results to be a more important source of uncertainties for our optimization as they cause interdependencies between the fitted parameters as well. This is especially relevant for the linear approach, where the functional dependencies of $CO_2$ and $O_2$ on the different river parameters are more similar than for the exponential approach (Fig. A2).

### D1 Parameter collinearity

The functional $CO_2$ dependency on $p$H, $O_2$, and DOC are more similar to each other for the linear than for the exponential $p$H approach (Fig. A2). This is also reflected in higher parameter uncertainties derived from the linear $p$H approach (Tab. 5). However, investigation of the correlation coefficients between the individual parameters reveal a strong positive correlation between the maximum decomposition rate ($R_{\max}$) and the Michaelis constant for $O_2$ ($K_m$) in both the linear and the exponential $p$H approach (Tab. D1). Additionally, there is a significant negative correlation between the exponential $p$H limitation constant ($\lambda$) and $K_m$ (Tab. D1).

**Table D1.** Correlations between the derived parameters.

| lin | $R_{\max}$ | $b$ | $K_m$ | exp | $R_{\max}$ | $b$ | $K_m$ | $\lambda$ |
|---|---|---|---|---|---|---|---|---|
| $R_{\max}$ | 1 | **−0.27** | **0.99** | $R_{\max}$ | 1 | **−0.29** | **0.82** | **−0.45** |
| $b$ | ∘ | 1 | **−0.25** | $b$ | ∘ | 1 | **−0.09** | **0.03** |
| $K_m$ | + | ∘ | 1 | $K_m$ | + | ∘ | 1 | **−0.86** |
| | | | | $\lambda$ | ∘ | ∘ | − | 1 |

Positive ($\rho \geq 0.5$ : +), negative ($\rho \leq -0.5$ : −) and non-significant ($-0.5 \leq \rho \leq 0.5$ : ∘) correlations between the parameters are indicated in the bottom left. The top right bold numbers represent the numerical Pearson correlation coefficients ($\rho$) between the parameters. The correlations are derived from the least-squares optimization of the linear $p$H approach (left table) and the exponential $p$H approach (right table). $R_{\max}$ is the maximum decomposition rate, $b$ is the fraction of $O_2$ consumption by decomposition, $K_m$ is the Michaelis constant for $O_2$ concentrations and $\lambda$ is the exponential $p$H limitation constant.

For the linear $p$H approach, the extremely high correlation between $R_{\max}$ and $K_m$ ($\rho = 0.99$) makes it impossible to meaningfully disentangle the individual impacts of these parameters. To test the possibility of a linear $p$H limitation in decomposition, least-squares correlations with fixed $K_m$ parameters within literature values ($1 - 40\,\mu\mathrm{mol L}^{-1}$, Fenoll et al., 2002) were performed (appendix D2).

For the exponential approach, while the parameters show a strong correlation ($\rho = 0.82$ for $R_{\max}$ & $K_m$ and $\rho = -0.86$ for $K_m$ & $\lambda$; Tab. D1), the functional dependencies are distinct enough to disentangle the parameter's impacts comparatively well

and the comparison to literature values supports the exponential $pH$ limitation. The high uncertainty in the $K_m$ parameter for this approach is only of small relevance as the $O_2$ limitation results to be comparatively weak. In fact, the $pH$ limitation alone is able to reproduce the measured parameters quite well (appendix D3).

## D2  Least-squares optimizations of linear $pH$ approach with fixed $K_m$ parameters

To test the possibility of a linear $pH$ limitation in decomposition, least-squares correlations for fixed $O_2$ Michaelis constants within literature values of $1-40\,\mu\mathrm{mol\,L^{-1}}$ (Fenoll et al., 2002) were performed (Tab. D2). The good agreement for all $K_m$ values is caused by the strong collinearity between $K_m$ and $R_{\mathrm{max}}$ that enables a change in $R_{\mathrm{max}}$ to compensate for changes in $K_m$. $R_{\mathrm{max}}$ values for the fixed $K_m$ values range between $1.4$ and $2.4\,\mu\mathrm{mol\,mol^{-1}\,s^{-1}}$. These maximum decomposition rates are lower than high decomposition rates derived based on global phenol oxidase activity by Sinsabaugh et al. (2008). The

fraction of $O_2$ consumption range between $102$ and $109\,\%$, indicating that for this approach to be true, more $O_2$ would need to be consumed than $CO_2$ is produced. In reality, additional $O_2$ can be taken from the organic matter, reducing the needed $O_2$ compared to $CO_2$ production. Thus, despite the good correlation to measured data (Tab. D2) the derived parameters for the linear approach do not agree well with literature data which makes this approach unlikely.

**Table D2.** Decomposition parameters for fixed $K_m$ values.

| $K_m$ | 1 | 2 | 4 | 6 | 10 | 15 | 20 | 30 | 40 | literature |
|---|---|---|---|---|---|---|---|---|---|---|
| $R_{\mathrm{max}}$ | $1.4\pm0.2$ | $1.5\pm0.2$ | $1.5\pm0.2$ | $1.6\pm0.2$ | $1.7\pm0.2$ | $1.8\pm0.2$ | $1.9\pm0.2$ | $2.1\pm0.2$ | $2.4\pm0.2$ | $\geq3.0$ |
| $b$ | $102\pm22$ | $103\pm23$ | $103\pm24$ | $104\pm25$ | $106\pm26$ | $108\pm28$ | $110\pm29$ | $110\pm29$ | $109\pm28$ | $\approx80$ |
| $R^2$ | 0.94 | 0.94 | 0.94 | 0.94 | 0.94 | 0.95 | 0.94 | 0.94 | 0.94 | - |
| RMSE | 182 | 181 | 180 | 179 | 178 | 177 | 176 | 174 | 172 | - |

Decomposition parameters derived for the linear $pH$ limitation approach via least-squares optimization of the equations in Tab. 1 with fixed Michaelis constant for $O_2$ concentrations given in $\mu\mathrm{mol\,L^{-1}}$. $R_{\mathrm{max}}$ is the maximum decomposition rate stated in $\mu\mathrm{mol\,mol^{-1}\,s^{-1}}$ and $b$ is the fraction of $O_2$ consumption by decomposition stated in %. Literature values for $R_{\mathrm{max}}$ and $b$ are taken from Sinsabaugh et al. (2008) and Rixen et al. (2008), respectively. Additionally, coefficients of determination ($R^2$) and root-mean-square errors (RMSE) of the correlation between measured $CO_2$ and $CO_2$ derived using these parameters and the equations in Tab. 1 are listed to indicate the quality of the fit.

## D3  Least-squares optimizations of exponential $pH$ approach without $O_2$ limitation

Our study revealed that low $pH$ is the main decomposition impelling parameter in peat-draining rivers. To study whether this parameter alone can explain the observed stagnation on $CO_2$ and $O_2$ for high peat coverage (Fig. 2), a least-squares optimization without $O_2$ limitation was performed. This optimization yields decomposition parameters that differ only insignificantly from the parameters derived for exponential $pH$ limitation with additional limitation by $O_2$ (Tab. D3). The correlation of measured $CO_2$ and $O_2$ to concentrations derived based on these parameters and the equations in Tab. 2 reveal a good agreement (Fig D1).

Only for the Kampar river, neglection of the $O_2$ limitation yields negative river $O_2$ concentrations (Fig D1). This indicates that

for $O_2$ concentrations in the examined rivers ($O_2 > 50\,\mu\mathrm{mol\,L^{-1}}$), the $p$H limitation alone is sufficient to explain the majority of the observed stagnation.

**Table D3.** Decomposition parameters for exponential $p$H limitation without $O_2$ limitation.

| parameter | with $O_2$ limitation | without $O_2$ limitation | unit |
|---|---|---|---|
| $R_{\mathrm{max}}$ | $4.0 \pm 0.8$ | $3.2 \pm 0.4$ | $\mu\mathrm{mol\,mol^{-1}\,s^{-1}}$ |
| $b$ | $81 \pm 10$ | $81 \pm 8$ | $\%$ |
| $K_m$ | $6 \pm 26$ | $0$ | $\mu\mathrm{mol\,L^{-1}}$ |
| $\lambda$ | $0.52 \pm 0.10$ | $0.54 \pm 0.05$ | |

Decomposition parameters derived for exponential $p$H limitation with and without additional $O_2$ limitation. Parameters were derived via least-squares optimization of the equations in Tab. 2 to measured data. $R_{\mathrm{max}}$ is the maximum decomposition rate, $b$ is the fraction of $O_2$ consumption by decomposition, $K_m$ is the Michaelis constant for $O_2$ limitation and $\lambda$ is the exponential $p$H limitation constant.

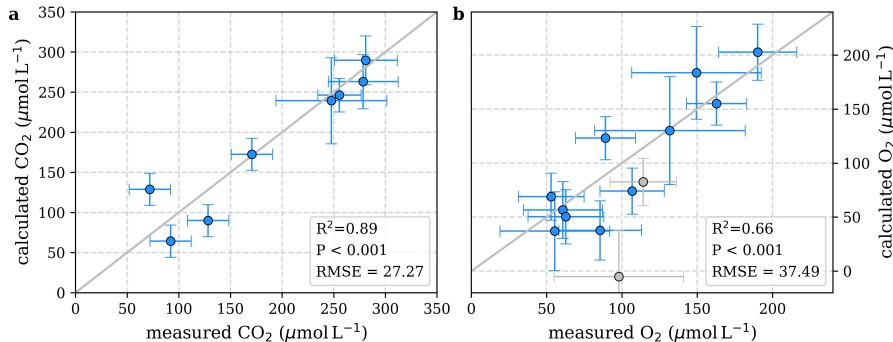

**Figure D1.** Correlation between measured and calculated concentrations of (a) $CO_2$ and (b) $O_2$. Grey lines indicate the 1:1 line. Calculations were performed for exponential $p$H limitation without $O_2$ limitation. Each data point represents one river. Grey data points are excluded from the correlation since the data for these rivers are based on less than three campaigns within the same season.

## D4 Validation of optimal $p$H for phenol oxidase activity

To validate the optimal $p$H for decomposition ($p\mathrm{H}_0$) in our study area, a least-squares optimization of the exponential $p$H approach (Tab. 2) including the parameter $p\mathrm{H}_0$ was performed. The resulting $p$H value of $p\mathrm{H}_0 \approx 7.2$ agrees well with the literature value of 7.5 used in our study (Tab. D4). However, it reveals a high collinearity to $R_{\mathrm{max}}$ that causes high parameter uncertainties.

**Table D4.** Decomposition parameters for exponential $pH$ limitation without $O_2$ limitation.

| parameter | fixed $pH_0$ | free $pH_0$ | unit |
|---|---|---|---|
| $pH_0$ | 7.5 | $7.2 \pm 153.1$ | |
| $R_{\mathrm{max}}$ | $4.0 \pm 0.8$ | $3.4 \pm 268.5$ | $\mathrm{\mu mol\,mol^{-1}\,s^{-1}}$ |
| $b$ | $81 \pm 10$ | $81 \pm 17$ | $\%$ |
| $K_m$ | $6 \pm 26$ | $6 \pm 29$ | $\mathrm{\mu mol\,L^{-1}}$ |
| $\lambda$ | $0.52 \pm 0.10$ | $0.52 \pm 0.11$ | |

Decomposition parameters derived for exponential $pH$ limitation with freely set $pH_0$
and with a fixed $pH_0$ of 7.5. Parameters were derived via least-squares optimization
of the equations in Tab. 2 to measured data. $R_{\mathrm{max}}$ is the maximum decomposition
rate, $b$ is the fraction of $O_2$ consumption by decomposition, $K_m$ is the Michaelis
constant for $O_2$ limitation and $\lambda$ is the exponential $pH$ limitation constant.

## D5 Decomposition approach for abnormal Simunjan campaigns

In the correlation figures Fig. 3 and 4, the Simunjan campaigns of January 2016 and March 2017 (Tab. 4) were excluded due to
scaling of the figures. Here we show the correlation figures with inclusion of those campaigns (Fig. D2 & D3). Since the data
is based on only one campaign, it was excluded from the least-squares optimization. Calculated $CO_2$ concentrations based on
both limitation approaches results to significantly higher concentrations than measured during the campaign (Fig. D2 & D3).
At the same time, calculated $O_2$ concentrations are lower than measured concentrations in the rivers.

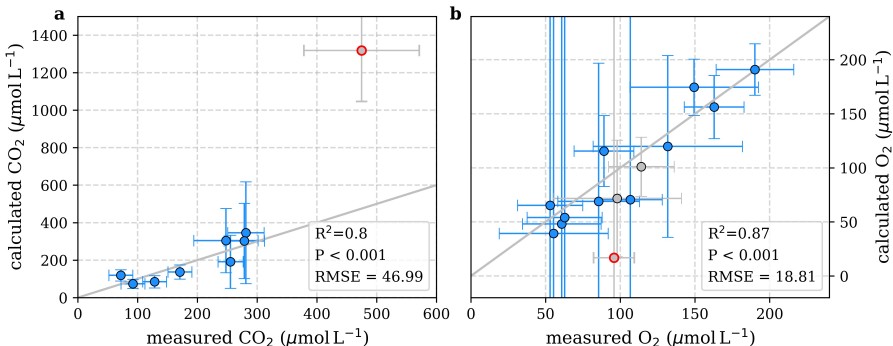

**Figure D2.** Correlation between measured and calculated concentrations of (a) $CO_2$ and (b) $O_2$. Grey lines indicate the 1:1 line. Calculations
were performed based on the equations in Tab. 1 which represent linear $pH$ limitation of decomposition rates. Each data point represents
one river. Grey data points are excluded from the correlation since the data for these rivers are based on less than three campaigns within the
same season. The Simunjan campaigns with high carbon concentrations (Simunjan$_2$, Tab. 4) are indicated by red circles.

This indicates that the parameters in these campaigns are not in equilibrium based on the processes of atmospheric gas exchange
and decomposition. This could be caused by additional processes of $CO_2$ sources and sinks during these anomalous campaigns.

However, since the observed events are temporal, we consider it likely that the river parameters simply had not reached a state of equilibrium yet. With such high carbon yields it is also possible that the river cannot reach a state of equilibrium before the water discharges into the ocean. However, as mentioned before, the data is mainly based on one campaign. To validate this assumption, further studies would be needed.

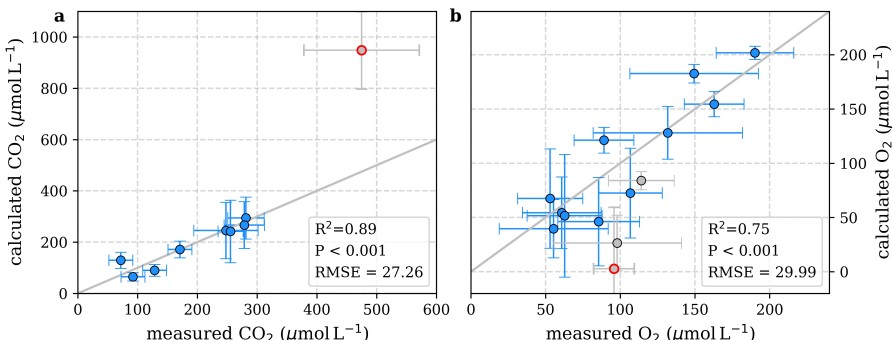

**Figure D3.** Correlation between measured and calculated concentrations of (a) $CO_2$ and (b) $O_2$. Grey lines indicate the 1:1 line. Calculations were performed based on the equations in Tab. 2 which represent exponential $p$H limitation of decomposition rates. Each data point represents one river. Grey data points are excluded from the correlation since the data for these rivers are based on less than three campaigns within the same season. The Simunjan campaigns with high carbon concentrations are excluded from these figures and further discussed in the appendix D5. The Simunjan campaigns with high carbon concentrations (Simunjan$_2$, Tab. 4) are indicated by red circles.

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
