# Peer review of "$CO_2$ emissions from peat-draining rivers regulated by water pH"

_Biogeosciences, 2021_

## Referee Comment (RC2)

**Major comments**

The contextualization and general justification of the paper could be revised. The authors justify their research to explain the discrepancy between estimates of $CO_2$ evasion by "global models" and those based on field measurements by their own group (for example Wit et al. 2015). The "global models" of Raymond et al. (2013) and Lauerwald et al. (2015) are not and not mechanistic models but in fact extrapolations of $pCO_2$ data calculated from pH and alkalinity measurements of unverified quality, that usually give results that are incorrect (Abril et al. 2015), and with a very coarse and extremely irregular spatial coverage. If you look at the maps of data point distribution of those two papers (in the supplements), for SE Asia there a handful of points in Thailand in the Raymond paper, and these data points did not meet the selection criteria of Lauerwald. In the Lauewarld paper that are in fact no data points at all for SE Asia.

In conclusion, the mismatch between field measurements and those predicted by Raymond et al. (2013) and Lauewarld et al. (2015) only shows that these "global models" are extremely unreliable, and does not reveal a hidden mechanism that lowers $CO_2$ emissions.

Conversely, the $pCO_2$ values reported for SE Asian peatland rivers, ranging between 2000 and 8000 ppm according to figure 2 of Wit et al. (2015) are within the range of $pCO_2$ reported in African tropical rivers (Borges et al. 2015) and also in rivers and streams of the Amazon River network (Abril et al. 2014). So the $pCO_2$ values in SE Asian peatland rivers seem relatively "normal" for tropical rivers, and not abnormally low.

The core topic of the paper is to look into the limitation of organic matter degradation (and subsequent $CO_2$ production) by low pH and low $O_2$. While it is intuitive that low $O_2$ and low pH might not be optimal to microbial growth, micro-organisms tend still to growth in sub-optimal conditions if there are substrates to metabolize. The correlations of $CO_2$ concentrations and pH/$O_2$ based on the data in Table 1 of the ms (see below) indicate on the contrary that the high $CO_2$ were associated to low pH and low $O_2$. And even if the conditions of pH and $O_2$ were sub-optimal, the micro-organisms were still able to degrade enough organic matter to produce large quantities of $CO_2$.

[Figure]

**Minor comments**

P1 L20: there could be a need to revise this statement in light of the work of Dargie et al. (2017).

**References**

Abril, G., Bouillon, S., Darchambeau, F., Teodoru, C. R., Marwick, T. R., Tamooh, F., Omengo, F. O., Geeraert, N., Deirmendjian, L., Polsenaere, P., and Borges A.V.: Technical note: Large overestimation of pCO2 calculated from pH and alkalinity in acidic, organic-rich freshwaters, Biogeosciences, 12, 67-78, doi: 10.5194/bg-12-67-2015, 2015.

Abril, G., Martinez, J.-M., Artigas, L.F., Moreira-Turcq, P., Benedetti, M.F., Vidal, L., Meziane, T., Kim, J.-H., Bernardes, M.C., Savoye, N., Deborde, J., Albéric, P., Souza, M.F.L., Souza, E.L., and Roland, F.: Amazon river carbon dioxide outgassing fuelled by wetlands, Nature, 505, 395-398, doi: 10.1038/nature12797, 2014.

Borges, A. V., Darchambeau, F., Teodoru, C. R., Marwick, T. R., Tamooh, F., Geeraert, N., Omengo, F. O., Guérin, F., Lambert, T., Morana, C., Okuku, E., and Bouillon, S.: Globally significant greenhouse-gas emissions from African inland waters, Nature Geoscience, https://doi.org/10.1038/ngeo2486, 2015.

Dargie et al. (2017) Age, extent and carbon storage of the central Congo Basin peatland complex, Nature 542, 86–90

Lauerwald, R., Laruelle, G. G., Hartmann, J., Ciais, P., and Regnier, P. A. G.: Spatial patterns in CO2 evasion from global river network, Global Biogeochemical Cycles, 29, 534–554, https://doi.org/10.1002/2014GB004941, 2015.

Raymond, P. A., Hartmann, J., Sobek, S., Hoover, M., McDonald, C., Butman, D., Striegel, R., Mayorga, E., Humborg, C., Kortelainen, P., Dürr, H., Meybeck, M., Ciais, P., and Guth, P.: Global carbon dioxide emissions from inland waters, Nature, 503, 355–359, https://doi.org/10.1038/nature12760, 2013.

Wit, F., Müller, D., Baum, A., Warneke, T., Pranowo, W. S., and Müller, M.: The impact of disturbed peatlands on river outgassing in Southeast Asia, Nature Communications, 6, https://doi.org/10.1038/ncomms10155, 2015.

---

## Author Response (AR1)

**Reply on Comments by Anonymous Referee #1:**

**Referee:**

The manuscript by Klemme et al. presents a study explaining why tropical peat draining rivers are only a moderate source of CO2 to the atmosphere, which stands in contrast to what was assumed for global estimates. Klemme et al. test the hypothesis that decomposition and thus CO2 production in these organic C rich waters is limited by pH and O2 availability. For this, they use a comprehensive dataset of observations of DOC and CO2 concentrations, pH and other relevant physical and chemical parameters from SE Asian, peat draining rivers in combination with a conceptual model representing limitations of DOC decomposition by low pH and O2 concentrations. They find that DOC decomposition in those peat draining rivers is likely more limited by pH than by O2, and suggest that increased loads of carbonates due to agricultural liming or enhanced weathering could increase decomposition of DOC and thus CO2 emissions from those peat draining rivers.

The study is original and of great interest for the readership of Biogeosciences. The manuscript is well written, the methodology is clearly described, and results are clearly presented and support the main findings of this study. I suggest publication after minor revisions. Please, find my comments below.

**Response:**

We thank the reviewer for the work with our manuscript and are pleased about their positive response to the concept and findings of our study. The suggestions by the reviewer were very helpful and improved the manuscript.

**Referee:**

L15-17 : Other studies have shown that large amounts of CO2 evading rivers are actually put in as dissolved CO2 from soil respiration (both heterotrophic and root respiration) (Abril and Borges, 2019; Lauerwald et al., 2020). Maybe you should mention that source as well.

**Response:**

We included this suggestion in our manuscript. In the revised manuscript we state: » ... *riverine CO2 is fed by decomposition of organic matter that is leached from soils (Wit et al., 2015) and by the leaching of dissolved CO2 from soil respiration (Abril and Borges, 2019; Lauerwald et al., 2020).* «

**Referee:**

L17-18: These are actually not model based studies that would represent peat soils. Those are more upscaling studies that lacked observations from these important systems

**Response:**

We realise that the use of the term "model-based studies" is imprecise. As the reviewer

points out these studies do not include soil models but are based on upscaling. We corrected this and in the updated manuscript we state: » *Despite scarcity in river CO2 measurements from Southeast Asia, studies suggest it as a hotspot for river CO2 emissions (Lauerwald et al., 2015; Raymond et al., 2013) due to the presence and degradation of carbon-rich peat soils.* «

**Referee:**
L42: In peat draining rivers, is there also less instream production by algae that would otherwise be a source of O2 to the water column?

**Response:**
Indeed, low nutrient concentrations (Baum and Rixen, 2014) as well as the dark water colour of peat-draining rivers that limit the light availability to algae (Wit et al., 2015) cause low rates of instream production. This further decreases O2 concentrations within those rivers that due to the high DOC and the concomitant high O2 consumption by decomposition exhibit low O2 concentrations. In the corrected manuscript we state: » *Due to high rates of decomposition caused by the carbon rich environment and low rates of photosynthesis caused by low nutrient concentrations and dark water colours that limit light availability to algae, peat-draining rivers are usually undersaturated with regard to atmospheric O2 (Wit et al., 2015, Baum and Rixen, 2014).* «

**Referee:**
L48-51: You should link these quite specific objectives here again to the more general research objective (or hypothesis to be tested): explain the moderate CO2 emissions from peat draining rivers by the effect of low pH and O2 limitation.

**Response:**
We included this as suggested. In the revised manuscript we state: » *This study aims at quantifying the impact of pH and O2 on the DOC decomposition in peat-draining rivers in order to explain the measured moderate CO2 emissions from those rivers by the limiting effect of these parameters.* «

**Referee:**
L95-97: I don't understand why you have used such a projection for determining areas. For that purpose, I would rather use an equal area projection, like an equal area projection after Lambert or the EckertIV projection.

**Response:**
Our phrasing at this point was misleading. We did not use the projection to determine catchment sizes but instead the Hydro-SHEDS data that our calculation of catchment sizes was based upon were provided in that geographical projection (Lehner et al., 2006). In the revised manuscript we rephrased this section to: » *Catchment sizes were derived from Hydro-SHEDS (Lehner et al., 2006) at 15s resolution in WGS 1984 Web Mercator Projection. Subbasins belonging to the catchments were identified using the HydroSHEDS 15s flow directions data set and added to the main basins.* «

**Referee:**
L110-112: The exponential limitation factor related to pH, which is defined as negative decadic logarithm of H+ activity - would that be comparable to a linear factor relating to the H+ activity? That might be worth discussing here in one or two sentences.

**Response:**
The logarithmic pH relation in the exponential limitation factor is indeed striking. Yet, it would not result in a linear correlation with the H+ activity but with this activity by the power of the exponential constant $\lambda$ divided by ln(10), which for the rivers we studied results to approximately 0.2. We included this information in the revised manuscript. In the methods we state: » *Considering the definition of pH as negative decadic logarithm of H+ activity ({H+}), the exponential limitation factor is equivalent to a correlation with {H+}^($\lambda$/ln(10)).* « and in the discussion we write: » *The exponential pH coefficient is $\lambda = 0.5 \pm 0.1$. Thus, in terms of H+ activity the correlation is given by {H+}^(0.5/ln(10)), which roughly equals the fifth root of {H+}.* «

**Referee:**
L122-123: That would require that dissolved CO2 inputs via groundwater inputs and CO2 consumption by autotrophic production is negligible. These are strong assumptions that would be worth mentioning here explicitly and some discussion later on.

**Response:**
We included this suggestion. In the methods we state: » *This approximation assumes photosynthetic CO2 consumption and direct CO2 input from leaching to be negligible.* « and later on we include discussions in form of: » *We acknowledge that this approximation assumes photosynthetic CO2 consumption and direct CO2 input from leaching to be negligible, which might not be the case for all rivers and we discuss the impact of these processes later on.* «
In the discussion we state: » *As mentioned before, our results do neglect the direct leaching of CO2 from soils and the consumption of CO2 by autotrophic production within the rivers. Since CO2 leaching rates are likely higher for peat soils than for mineral soils (Kang et al., 2018) and autotropic production is limited in peat-draining rivers (Wit et al., 2015), both of these processes would work against the observed recession in CO2 growth. This indicates that exclusion of those processes could cause underestimation of the limitation factors rather than overestimation.* «

**Referee:**
L140: "spatially as well as temporally"

**Response:**
*We changed this as suggested.*

**Referee:**
Figure 3: The grey lines, are those regression fits or the 1:1 line, or both?

**Response:**
*Those lines represent the 1:1 line. We included this information in the revised manuscript.*

**Referee:**

For figures 3 and 4, it would be great if you could report in addition the RMSEs.

**Response:**

*We included those as suggested.*

**Referee:**

L184: There's a "c" missing in "concentration".

**Response:**

*We changed this.*

**Referee:**

L189-191: Do Borges et al. also report CO2 emission rates or CO2 concentrations which are comparable to those in your study?

**Response:**

Yes, the CO2 and DOC concentrations by Borges et al are comparable to the concentrations measured in our study. In the revised manuscript we included this information and state: *» A similar pattern of stagnating CO2 concentrations has been observed in river sections of high DOC at the Congo river (Borges et al., 2015). The CO2 and DOC concentrations measured in these rivers are comparable to those measured in our study, indicating that the underlying process is valid not only for Southeast Asian rivers but for tropical peat-draining rivers in general. «*

**Reply on Comments by Anonymous Referee #2:**

**Referee:**

The contextualization and general justification of the paper could be revised. The authors justify their research to explain the discrepancy between estimates of CO2 evasion by "global models" and those based on field measurements by their own group (for example Wit et al. 2015). The "global models" of Raymond et al. (2013) and Lauerwald et al. (2015) are not and not mechanistic models but in fact extrapolations of pCO2 data calculated from pH and alkalinity measurements of unverified quality, that usually give results that are incorrect (Abril et al. 2015), and with a very coarse and extremely irregular spatial coverage. If you look at the maps of data point distribution of those two papers (in the supplements), for SE Asia there a handful of points in Thailand in the Raymond paper, and these data points did not meet the selection criteria of Lauerwald. In the Lauewarld paper that are in fact no data points at all for SE Asia.

**Response:**

We thank the reviewer for the work with the manuscript. The use of the term "model-based" was also criticised by the first reviewer and we changed the statement to: *» Despite scarcity in river CO2 measurements from Southeast Asia, studies suggest it as a*

*hotspot for river CO2 emissions (Lauerwald et al., 2015; Raymond et al., 2013) due to the presence and degradation of carbon-rich peat soils. «*

The reviewer is right in that the mismatch between those studies and measured data is not surprising considering the data scarcity and consequential uncertainties. Furthermore, the results by Lauerwald et al. (2015) are within the range of measured CO2 concentrations (Wit at al., 2015). Thus, according to the reviewer's suggestion, we shift the motivation of our study from discrepancies between those upscaling studies and measurements towards the surprisingly low CO2 measurements in rivers of high DOC concentrations. In the revised manuscript we state: » *However, despite high leaching rates that cause DOC concentration which can be more than four times higher than those in temperate regions (Butman and Raymnond, 2011; Müller et al., 2015), measured CO2 fluxes from tropical peat-draining rivers (25.2 gC m$^{-2}$ yr$^{-1}$) hardly exceed those measured for rivers in temperate regions (18.5 gC m$^{-2}$ yr$^{-1}$; Butman and Raymond, 2011; Wit et al., 2015). «*

**Referee:**

In conclusion, the mismatch between field measurements and those predicted by Raymond et al. (2013) and Lauewarld et al. (2015) only shows that these "global models" are extremely unreliable, and does not reveal a hidden mechanism that lowers CO2 emissions.

**Response:**

As we state above, it is not only the mismatch between these specific studies and measurements that we want to explain. The question we want to answer is the cause for the rather moderate CO2 concentrations measured in tropical peat-draining rivers given the high DOC concentrations. The presence of carbon-rich peat soils and consequently high concentrations of DOC in peat-draining rivers should result in high CO2 concentrations, but the measurements show that the CO2 concentrations in these rivers are only insignificantly higher than emissions from temperate regions (Wit et al., 2015). In this study, we aim at explaining the process that is limiting the CO2 production given the high DOC concentrations.

**Referee:**

Conversely, the pCO2 values reported for SE Asian peatland rivers, ranging between 2000 and 8000 ppm according to figure 2 of Wit et al. (2015) are within the range of pCO2 reported in African tropical rivers (Borges et al. 2015) and also in rivers and streams of the Amazon River network (Abril et al. 2014). So the pCO2 values in SE Asian peatland rivers seem relatively "normal" for tropical rivers, and not abnormally low.

**Response:**

This is exactly what we want to explain. Despite high DOC concentrations in tropical rivers, the CO2 emissions are relatively moderate (Wit et al., 2015). It is not the aim of our study to discuss or explain that CO2 values in Southeast Asian rivers are abnormally low in contrast to other tropical rivers. What we refer to when mentioning

moderate CO2 is the stagnating concentration in rivers of high peat coverage, in which DOC concentrations are extremely high. Yet, while DOC concentrations can be by a factor 5 higher than in temperate regions, CO2 emissions are not even twice as high as those stated for temperate rivers. Similarly, for the rivers in our study, CO2 concentrations do not change significantly for rivers with DOC concentrations between 2,000 and 4000 µmol L$^{-1}$. Since our dataset is specifically based on measurement campaigns in Southeast Asian rivers, we focus on quantifying the limitation factors for rivers within this area. However, we explicitly state that a similar limitation is likely present in tropical peat-draining rivers in general (line 190-191). According to a suggestion of reviewer #1, this section was adjusted to: *» A similar pattern of stagnating CO2 concentrations has been observed in river sections of high DOC at the Congo river (Borges et al., 2015). The CO2 and DOC concentrations measured in these rivers are comparable to those measured in our study, indicating that the underlying process is valid not only for Southeast Asian rivers but for tropical peat-draining rivers in general. «*

**Referee:**

The core topic of the paper is to look into the limitation of organic matter degradation (and subsequent CO2 production) by low pH and low O2. While it is intuitive that low O2 and low pH might not be optimal to microbial growth, micro-organisms tend still to growth in sub-optimal conditions if there are substrates to metabolize. The correlations of CO2 concentrations and pH/O2 based on the data in Table 1 of the ms (see below) indicate on the contrary that the high CO2 were associated to low pH and low O2. And even if the conditions of pH and O2 were sub-optimal, the micro-organisms were still able to degrade enough organic matter to produce large quantities of CO2.

**Response:**

The correlation of CO2 concentrations and pH/O2 pointed out by the reviewer neglects the DOC concentration. The rivers of low pH and low O2 are the rivers with a high peat coverage in the catchment and therefore high DOC. Obviously, the CO2 concentration is high in these rivers, since the DOC decomposition increases with the amount of available DOC in the rivers. This is why we define the decomposition rate as produced CO2 per available DOC. If this decomposition rate was not limited by parameters like pH or O2, the CO2 production would linearly increase with DOC concentrations, which would result in a fairly linear increase of CO2 concentrations with DOC and therewith significantly higher CO2 concentrations than observed in peat-draining rivers.

**Referee:**

Minor comments P1 L20: there could be a need to revise this statement in light of the work of Dargie et al. (2017).

**Response:**

It is a good suggestion to include the study by Dargie et al. in our manuscript. Their results increase the total tropical peat carbon store from approximately 89 PgC to 105 PgC. Southeast Asian peatlands are estimated to store more than 60 PgC (Page et al, 2011). Thus, these data conform with our original statement that more than half of the known tropical peatlands are located in Southeast Asia. In the revised manuscript we state: » *More than half of the known tropical peatlands are located in Southeast Asia (Dargie et al., 2017; Page et al., 2011), whereat 84 % of these are Indonesian peatlands, mainly on the islands of Sumatra, Borneo and Irian Jaya (Page et al., 2011).* «

**References**

**Abril et al. 2014:** Abril, G., Martinez, JM., Artigas, L. et al. Amazon River carbon dioxide outgassing fuelled by wetlands. Nature 505, 395–398 (2014).

**Abril et al., 2015:** Abril, G., Bouillon, S., Darchambeau, F., Teodoru, C. R., Marwick, T. R., Tamooh, F., Ochieng Omengo, F., Geeraert, N., Deirmendjian, L., Polsenaere, P., and Borges, A. V.: Technical Note: Large overestimation of pCO2 calculated from pH and alkalinity in acidic, organic-rich freshwaters. Biogeosciences 12, 67–78 (2015).

**Abril and Borges, 2019:** Abril, G. and Borges, A. V.: Ideas and perspectives: Carbon leaks from flooded land: do we need to replumb the inland water active pipe?. *Biogeosciences* 16, 769–754 (2019).

**Baum and Rixen, 2014:** Baum, A. and Rixen, T.: Dissolved inorganic nitrogen and phosphate in the human affected blackwater river Siak, central Sumatra, Indonesia. *Environment and Pollution* 11, 13-24 (2014).

**Borges et al., 2015:** Borges, A., Darchambeau, F., Teodoru, C. et al.: Globally significant greenhouse-gas emissions from African inland waters. Nature Geoscience 8, 637–642 (2015).

**Butman and Raymnond, 2011:** Butman, D. and Raymond, P.: Significant efflux of carbon dioxide from streams and rivers in the United States. Nature Geoscience 4 (2011).

**Dargie et al., 2017:** Dargie, G., Lewis, S., Lawson, I. et al.: Age, extent and carbon storage of the central Congo Basin peatland complex. Nature 542, 86–90 (2017).

**Lauerwald et al., 2015:** Lauerwald, R., Laruelle, G. G., Hartmann, J., Ciais, P., and Regnier, P. A.: Spatial patterns in CO2 evasion from the global river network. Global Biogeochem. Cycles 29, 534– 554 (2015).

**Lauerwald et al., 2020:** Lauerwald, R., Regnier, P., Guenet, B., Friedlingstein, P. and Ciais, P.: How Simulations of the Land Carbon Sink Are Biased by Ignoring Fluvial Carbon Transfers: A Case Study for the Amazon Basin. *One Earth* 3, 226-236 (2020).

**Lehner et al., 2006:** Lehner, B., Verdin, K., and Jarvis, A.: HydroSHEDS, Technical Documentation. Tech. rep., HydroSHEDS, version 1.0, 1-27, (2006).

**Müller et al., 2015:** Müller, D., Warneke, T., Rixen, T., Müller, M., Jamahari, S., Denis, N., Mujahid, A., and Notholt, J.: Lateral carbon fluxes and CO2 outgassing from a tropical peat-draining river. Biogeosciences 12, 5967–5979 (2015).

**Page et al., 2011:** Page, S. E., Rieley, J. O., and Banks, C. J.: Global and regional importance of the tropical peatland carbon pool. Global Change Biology 17, 798–818 (2011).

**Raymond et al., 2013:** Raymond, P. A., Hartmann, J., Sobek, S., Hoover, M., McDonald, C., Butman, D., Striegel, R., Mayorga, E., Humborg, C., Kortelainen, P., Dürr, H., Meybeck, M., Ciais, P., and Guth, P.: Global carbon dioxide emissions from inland waters. Nature 503, 355–359 (2013).

**Wit et al., 2015:** Wit, F., Müller, D., Baum, A., Warneke, T., Pranowo, W. S., and Müller, M.: The impact of disturbed peatlands on river outgassing in Southeast Asia. Nature Communications 6, (2015).

---

## Author Response (AR2)

**Reply on Comments by Anonymous Referee #3:**

**Referee:**

This manuscript explores the controls on CO2 emissions from peat-draining rivers, finding that pH limitation plays a central role. It is an important regional-scale analysis addressing a very interesting and understudied topic, which is relevant to the readership of Biogeosciences. However, the manuscript still requires major revisions to ensure the central findings are clearly documented and sufficient uncertainty analysis is provided for the readers. Additionally, further updates to writing and references would strengthen the paper.

**Response:**

We thank the reviewer for their thorough work with our manuscript and are pleased about their positive response to the study topic and concept. Implementation of their suggestions certainly improved our manuscript.

**Referee:**

Provide additional information on methods, fitted parameters, and implications for results: In the current manuscript, the main findings are not sufficiently documented, leaving the reader with substantial uncertainties related to the approach and conclusions. The central conclusion that pH limitation dominates hinges on the values of the fitted parameters. Could you please provide supplemental figures to provide more insight into the methods and uncertainty analysis?

For example, in equations in Table 2 & 3, CO2 concentration will be very insensitive to O2 concentration for small Km, but will become more sensitive for higher Km. Fitted values of Km varied widely (factor of 50) between the two model formulations. The authors used this variation to rule out the linear approach in favour of the exponential approach. Please provide more information to justify the case for this interpretation. Some possible questions and ideas are below, but other information and analyses would also be welcome.

**Response:**

We revised the methods and included supplementary information and additional figures to make our process more transparent.

Table 01 summarizes the parameters resulting from different decomposition approaches. All of those approaches, with correlation coefficients of 0.75 to 0.95, yield good correlations to measured CO2 and O2 concentrations. However, the associated decomposition parameters (maximum decomposition rate (Rmax), fraction of O2 consumption (b), Michaelis constant for O2 limitation (Km) and exponential pH limitation factor ( $\lambda$ )) differ between the approaches.

For the exponential pH limitation approach, all derived parameters agree with values stated in literature (Table 01). For the linear pH approach, the least-squares optimization yields unrealistically high Km values. These high Km values are partially caused by a strong collinearity between Rmax and Km in the linear approach (that we included in appendix D1 and discuss later in this response). However, an additional least-squares optimization with a fixed Km constant of 20 µmol/L (based on data stated by Fenoll et al. (2002)) yields parameters for Rmax and b that disagree with literature values (Table 01). Thus, in the manuscript we now write:

Our results indicate the exponential pH limitation of decomposition to be more realistic than the linear pH limitation. The exponential limitation better represents river CO2 especially for high CO2 concentrations which are most strongly affected by the pH limitation. The exponential limitation is additionally supported by the unrealistically high O2 limitation resulting from the linear pH approach. The strong collinearity between decomposition parameters in the linear pH limitation approach complicates the interpretation of the parameters mentioned above. Additional calculations of the parameters Rmax and b for fixed Km values also disagree with literature data and thus further disprove the linear approach (appendix D2).

| parameter            | exp. pH
approach | lin. pH
approach | Lin. pH
approach
with
fixed Km | Only O2
limitation
with
fixed Km | Only pH
limitation | Literature
values | unit       |
|----------------------|---------------------|---------------------|-----------------------------------------|-------------------------------------------|-----------------------|----------------------|------------|
| Km                   | $6 \pm 26$          | $390 \pm 508$       | 20                                      | 20                                        | 0                     | $1 - 40^{[1]}$       | μmol/L     |
| Rmax                 | $4.0 \pm 0.8$       | $10 \pm 11$         | $1.9 \pm 0.2$                           | $1.0 \pm 0.7$                             | $3.3 \pm 0.4$         | $\geq 3^{[2]}$       | µmol/mol/s |
| b                    | $81 \pm 10$         | $90 \pm 25$         | $110\pm29$                              | $174 \pm 78$                              | $81 \pm 8$            | $\approx 80^{[3]}$   | %          |
| λ                    | $0.5 \pm 0.1$       | -                   | -                                       | 0                                         | $0.5 \pm 0.1$         | $0.6 - 0.8^{[4]}$    | -          |
| R 2 (CO2) | 0.89                | 0.80                | 0.94                                    | 0.76                                      | 0.89                  | -                    | -          |
| R 2 (O2)  | 0.86                | 0.87                | 0.84                                    | 0.84                                      | 0.86                  | -                    | -          |

**Table 01:** Parameters derived from least-squares approximations to measured data. Km is the Michaelis constant for O2, Rmax is the maximum decomposition rate, b is the fraction of O2 consumption by decomposition and  $\mathbb{R}^2$  is the coefficient of determination that provides an indicator for the fit's quality. Values that were set fixed rather than derived via the least-squares optimization are indicated by bold grey numbers. Literature values for this comparison were taken from [1] Fenoll *et al.* (2002), [2] Sinsabaugh *et al.* (2008), [3] Rixen *et al.* (2008) and [4] Williams *et al.* (2000).

The dominance of pH limitation over O2 limitation results directly from these fitted parameters, since mathematically the O2 and pH limitations can be derived as functions of O2 & Km and of pH and  $\lambda$ , respectively. For the exponential pH limitation, these functions result to a limiting impact of < 10 % for O2 and of up to 85 % for pH. In fact, a least-squares optimization of the exponential pH approach excluding O2 limitation revealed that the pH limitation alone is able to explain the majority of the observed stagnation in CO2 and O2 concentrations (Table 01). This additional least-squares optimization is included in appendix D3 and will be further discussed later in this response.

In the course of a more detailed quality assessment of the least-squares optimizations, we also performed a more detailed examination of the optimum pH value for decomposition (pH0, appendix D4) as well as a dedicated discussion of the abnormal Simunjan campaigns that were excluded from the least-squares optimization (appendix D5). In appendix D4, we now added:

To validate the optimal pH for decomposition (pH0) in our study area, a least-squares optimization of the exponential pH approach (Tab. 2) including the parameter pH0 was performed. The resulting value of pH0  $\approx$  7.2 agrees well with the literature value of 7.5 used in our study (Tab. D4). However, it reveals a high collinearity to Rmax that causes high parameter uncertainties.

And:

In the correlation figures Fig. 3 and 4, the Simunjan campaigns of January 2016 and March 2017 (Tab. 4) were excluded due to scaling of the figures. Here we show the correlation figures with inclusion of those campaigns (Fig. D2 & D3). Calculated CO2

concentrations based on both limitation approaches results to significantly higher concentrations than measured during the campaign. At the same time, calculated O2 concentrations are lower than measured concentrations in the rivers.

This indicates that the parameters in these campaigns are not in equilibrium based on the processes of atmospheric gas exchange and decomposition. This could be caused by additional processes of CO2 sources and sinks during these anomalous campaigns. However, since the observed events are temporal, we consider it likely that the river parameters simply had not reached a state of equilibrium yet. With such high carbon yields it is also possible that the river cannot reach a state of equilibrium before the water discharges into the ocean. However, as mentioned before, the data is mainly based on one campaign. To validate our assumption, further studies would be needed.

In the following, we explain the changes to our manuscript in more detail based on point-topoint answers on the reviewer's comments.

**Referee:**

**Questions/Ideas---**

-Table 5: Could you provide a related figure showing the least squares optimization and/or model fits with these parameters? I only see the predicted vs. observed, so more information would be useful.

**Response:**

The correlations in Fig. 2 & 3 show the quality of the fits. The equations used for the least-squares optimizations depend on various measured parameters (O2, pH, DOC & T) that differ for the investigated rivers. Therefore, a visualisation of all dependencies is in our opinion not possible.

**Referee:**

-Table 5 and Table 6: Could you provide the fitted parameters, and pH and O2 limitations for both the linear and exponential formulations so we can see how they differ? Both appear to have good performance in Figure 3 & 4, so it would be interesting for the reader to see both propagated throughout the manuscript. Very confusing when they are contrasted in the discussion, but the linear parameter fit values cannot be viewed in any table.

**Response:**

We included the fit parameters for the linear approach as suggested by the reviewer. Those factors are listed alongside the exponential factors in Tab. 5. The limitation factors for the linear and the exponential approach have been re-located to Tab. A3 and Tab. A4 in the appendix, respectively.

**Referee:**

-Minor formatting: inconsistent ordering of linear and exponential is confusing (Table 2&Figure 4; Table 3 & Fig 3). Linear is missing from later tables.

**Response:**

We changed the order of the tables in the methods section such that the linear approach is always mentioned first and included tables for the linear correlation as mentioned.

-Could you provide more insight into why the fitted values were so different for the two formulations?

**Response:**

The parameter that varies the strongest between the linear and the exponential approach is the Michaelis constant for O2 (Km). This parameter represents the O2 concentration at which decomposition is limited by 50%. The exponential and linear least-squares approximations yield very differing results and the Km resulting from the linear approach is unrealistically high (Table 01).

Our analysis indicates that this high Km for the linear pH limitation approach is caused by a strong collinearity between Km and the maximum decomposition rate (Rmax). We included a discussion of these parameter collinearities in the manuscript's appendix D1, where we state:

»The functional CO2 dependency on pH, O2, and DOC are more similar to each other for the linear than for the exponential pH approach (Fig. A2). This is also reflected in higher parameter uncertainties derived from the linear pH approach (Tab. 5). [...] For the linear pH approach, the extremely high correlation between Rmax and Km ( $R^2 = 0.99$ ) makes it impossible to meaningfully disentangle the individual impacts of these parameters. To test the possibility of a linear pH limitation in decomposition, least-squares optimizations with fixed Km parameters within literature values (1 – 40 µmol L-1, Fenoll et al., 2002) were performed (appendix D2).«

These optimizations yield lower Rmax values and higher b values than stated by literature (Table 01). In the appendix D2 we therefore conclude that:

[...] despite the good correlation to measured data (Tab. D2) the derived parameters for the linear approach do not agree well with literature data which makes this approach unlikely.

In the conclusions of the main manuscript, we state:

[...] The linear pH limitation approach yields a Michaelis constant of  $\text{Km} \approx 390 \text{ }\mu\text{mol} \text{ }L^{-1}$ . This constant is higher than the O2 concentration in atmospheric equilibrium ( $\approx 280 \ \mu mol \ L^{-1}$ ), which implies an oxygen deficit at atmospheric conditions that does not exist (Vaquer-Sunyer and Duarte, 2008). However, though the derived Km value for this linear pH limitation is unrealistically high, this does not necessarily negate the linear pH approach. High parameter interdependence between Km and Rmax complicate the computation of these decomposition parameters (appendix D1). To disentangle the impact of the intercorrelated parameters, additional least-squares optimizations at fixed Km values ranging from 1 to 40 µmol L-1 (Fenoll et al., 2002) were performed (appendix D2). These optimizations result in maximum decomposition rates of Rmax = (1.4 - 2.4) µmol mol-1 s-1 and O2 consumption factors of b = (102 - 109) % and therewith do not agree with literature values of these parameters  $(\text{Rmax} \ge 3 \ \mu\text{mol mol}^{-1} \ \text{s}^{-1} \ \& \ b \approx 80 \ \%;$  Sinsabaugh et al., 2008; Rixen et al., 2008). [...]

**Referee:**

-Suggested figure: Plot of key equations from Table 2 and Table 3 shown with data used for fitting

**Response:**

A suitable visualization of the equations in Table 2 and Table 3 for the measured data is in our view not possible, because the equations depend on four parameters that all differ for the investigated rivers. However, to illustrate the dependencies of CO2 and O2, we have included figures illustrating the individual dependencies in the appendix as Fig. A2 and Fig. A3.

**Referee:**

-Suggested SI figure: Plot of key equations from Table 2 and Table 3 with different values of fitted parameters to give readers an idea of sensitivity to these parameters.

**Response:**

We included such figures in the supplement as Fig. A4 to Fig. A7. The figures show the variation in CO2 and O2 concentrations derived for the individual rivers based on variation in the different fitted parameters. All parameters are varied within the derived uncertainties ( $1\sigma$  value derived from least-squares optimization). For comparison, the figures also include the measured and average calculated CO2 and O2 values.

At the end of this document, we additionally included Figures that combine Fig. A2 (CO2 dependency on measured parameters) with the figures A4 & A5 (CO2 sensitivity on fitted parameters). The Additional Figures 1&2 show the dependencies for the linear approach and the Additional Figures 3-5 show the dependencies for the exponential approach. However, we decided not to include these figures in the manuscript, as they provide only minor additional information compared to the figures A2, A4 & A5.

**Referee:**

-How confident are you in your ability to disentangle pH and O2 effects given the noise in the data? Can you help the reader understand how different the table 2 &3 equation curves would look with different combinations of parameters? Are they similar or strongly distinguishable?

**Response:**

The linear and the exponential pH limitation approaches represent limitation of mainly O2 (linear approach) and of mainly pH (exponential approach). Both approaches reveal strong interdependencies between the fitted parameters. We include a discussion of this in the appendix D. We state:

Uncertainty sources in the least-squares optimizations are interdependencies between the fitted parameters and noise in the measured data. We try to minimize the impact of measurement noise by including relative uncertainties (o) of measured CO2 and O2 concentrations in the least-squares optimization. Thus, data from rivers with higher variation in measured parameters are constrained less rigidly in the optimization. The parameter interdependence results to be a more important source of uncertainties for our optimization fit, as they cause interdependencies between the fitted parameters as well. This is especially relevant for the linear approach, where the functional dependencies of CO2 and O2 on the different river parameters are more similar than for the exponential approach (Fig. A2).

In the appendix D1, we discuss the parameter collinearities in more detail:

The functional CO2 dependency on pH, O2, and DOC are more similar to each other for the linear than for the exponential pH approach (Fig. A2). This is also reflected in

higher parameter uncertainties derived from the linear pH approach (Tab. 5). However, investigation of the correlation coefficients between the individual parameters reveals a strong positive correlation between the maximum decomposition rate (Rmax) and the Michaelis constant for O2 (Km) in both the linear and the exponential pH approach (Tab. D1). Additionally, there is a significant negative correlation between the exponential pH limitation constant ( $\lambda$ ) and Km (Tab. D1).

As mentioned before, the strong collinearity in the linear approach makes it impossible to meaningfully disentangle the different dependencies. This is why the additional least-squares optimizations in appendix D2 were performed. For the exponential pH approach, however, we are confident that the derived parameter dependencies are meaningful despite the parameter intercorrelation. In the appendix D1 we state:

For the exponential approach, while the parameters show a strong correlation  $(R^2 = 0.82 \text{ for Rmax \& Km} \text{ and } R^2 = 0.86 \text{ for Km \& } \lambda; \text{ Tab. D1})$ , the functional dependencies are distinct enough to disentangle the parameter's impacts comparatively well and the comparison to literature values supports the exponential pH limitation. The high uncertainty in the Km parameter for this approach is only of small relevance as the O2 limitation results to be comparatively weak. In fact, the pH limitation alone is able to reproduce the measured parameters quite well (appendix D3).

**Referee:**

-Suggested figure: any assessment of relationships or collinearity between fitted parameters

**Response:**

We included a discussion of the parameter's collinearity based on correlation coefficients in the appendix D1. This section includes a table that lists collinearities between the fitted parameters. In the end of this document, we additionally included figures of the parameter's dependencies on each other (Additional Figures 6 & 7). However, in our opinion the table is sufficient for the scope of the manuscript.

**Referee:**

-If O2 limitation is negligible, why is there such a strong inverse trend with peat cover vs. O2 & CO2? (Figure 2)

**Response:**

Much like the CO2-peat coverage trend, the O2-peat coverage trend is caused implicitly by the DOC-peat coverage correlation and the pH-peat coverage correlation. The increase in DOC with peat coverage causes increased decomposition and thus CO2 production and O2 consumption. The co-occurring decrease in pH causes the stagnation in both CO2 and O2 concentrations for high peat coverage due to its limiting impact on decomposition.

**Referee:**

-Could you add a panel for pH in Figure 2, given the central role of pH?

**Response:**

Fig. 2b shows the pH-peat coverage correlation.

**Referee:**

-What is the significance of the exponential correlation lines shown in Figure 2? Do they have

any relationship to the equations in other parts of the manuscript, or any other interpretation or analysis?

**Response:**

The exponential correlations were included solely to show the nonlinearity between the parameters. There is no scientific relationship to other equations in the manuscript, as they refer to CO2 – peat coverage and O2 – peat coverage correlations that are implicitly driven by the parameter dependencies on DOC and pH.

**Referee:**

-Table 6: How are these limitations and uncertainties calculated? Please add more info in Methods.

**Response:**

We included this information in the methods. After we introduce the limitation factors  $L_{pH}$  and  $L_{02}$ , we state:

[...] The limitation factors represent the fraction of decomposition that is remaining after the limitation by the parameter. Later on, we refer to the fraction by which decomposition is limited, which is  $(1 - L_{pH})$  for pH limitation and  $(1 - L_{02})$  for O2 limitation. The total fraction by which pH and O2 limit decomposition is given by  $(1 - L_{pH} \cdot L_{02})$ . [...]

**Referee:**

-Ln 90- Provide equations used from Wannikhof (1992) within your methods to make your work easier to reproduce. Almost all other parameters needed to use eqns in Table 2 & 3 are already provided in Table 1, so please include kCO2(T) or equation to calculate it for completeness.

**Response:**

We included the equations from Wanninkhof (1992) for CO2 and O2 exchange coefficients  $(k_CO2(T) \& k_O2(T))$  in the methods section. For completeness, we additionally included the equation for the Henry coefficients of CO2 and O2 as derived by Weiss (1972) and Weiss (1970). Additionally, parameters derived from these equations for the individual rivers are included in Tab. A2 in the appendix.

**Referee:**

-Ln 218 – If fitted parameters imply no oxygen limitation, is this still able to fit data from Fig 2, or are some extremes missed entirely?

**Response:**

To answer this question, we performed an additional least-squares optimization without O2 limitation. We show the results of this optimization in appendix D3. We find that:

[...] This optimization yields decomposition parameters that differ only insignificantly from the parameters derived for exponential pH limitation with additional limitation by O2 (Tab. D3). The correlation of measured CO2 and O2 to concentrations derived based on these parameters and the equations in Tab. 3 reveal a good agreement (Fig D1). Only for the Kampar River, neglection of the O2 limitation yields negative river O2 concentrations (Fig D1). This indicates that for O2 concentrations in the examined rivers (O2 > 50  $\mu$ mol L-1), the pH limitation alone is sufficient to explain the majority of the observed stagnation.

-----Other concepts requiring further discussion:

-Photomineralization -- Recent literature has suggested that CO2 emissions from peat-draining rivers may be largely driven by photomineralization, rather than microbial respiration. Would this process be expected to have the same formulation of oxygen or pH limitations? What similarities or differences would you expect? How might this change the equations you used? Would there be a strong theoretical basis for pH or oxygen limitation of photomineralization? A paragraph reviewing this issue would be helpful.

**Response:**

We included a discussion of the effect of photomineralization to our manuscript. In the conclusions we state:

A recent study by Nichols and Martin (2021) found low phenol oxidase activity in Southeast Asian peat-draining rivers and low degradation of DOC from those rivers in an additional incubation experiment. They concluded that that the remineralization of peat-derived DOC in Southeast Asian aquatic systems is likely dependent on photodegradation rather than microbial respiration (Nichols and Martin, 2021). This is supported by photolability of DOC from those regions (Martin et al., 2018). However, photomineralization rates would not be impacted by river pH or O2. Thus, with photomineralization as the main cause of DOC degradation, no stagnation in CO2 is expected. Accordingly, photomineralization of DOC, like the before-mentioned processes, would work against the observed CO2 stagnation and could cause underestimation of the limitation parameters.

**Referee:**

-DOM composition - Ln 31 (also 36): You make the contrast with temperate peatlands. Could you also comment somewhere on the differences between the peat and DOM composition in temperate vs. tropical peatlands, and how that might be another factor slowing decomposition?

For example, see work by Hogkins et al (2018). Also Nichols & Martin (2021).

**Response:**

The most important difference between temperate and tropical peat soils is the peat composition. Tropical peat soils generally contain higher fractions of phenolic compounds than temperate peat soils. Therefore, the presence and activity of phenol oxidase is especially important in our tropical study area. In our manuscript introduction we state:

[...] Phenol oxidase is needed to decompose phenolic compounds that are especially present in tropical peat soils (Hodgkins et al., 2018; Yule et al., 2018). [...]

**Referee:**

-Carbonates:

-Ln 282: Mention and discussion of enhanced weathering is interesting. However, it does not make sense to me to have this as the final concluding paragraph, as it is not the central message of the manuscript. Perhaps it could be relocated?

**Response:**

We restructured the conclusions such that it finishes with a final paragraph on the global

relevance of pH as regulator of CO2 emissions. The paragraph about enhanced weathering was removed from the conclusions and replaced by a short mention, as more detailed implications for it were already provided in the discussions.

In this manuscripts' conclusion we state:

[...] Possible sources for enhanced carbonate concentrations can be soil erosion upstream of coastal peatland areas, or liming practices in plantations along the rivers, which are common practice to improve plant growth on acidic soils. This carbonate impact should be considered for anthropogenic activities like liming and enhanced weathering.

Our study is based on measurements in Southeast Asian peat-draining rivers. However, comparison to data from African rivers and laboratory studies of decomposition in temperate peat soils suggest that the investigated correlations and processes are also relevant in other regions and that soil and water pH are important regulators of global carbon emissions.

**Referee:**

-Ln 276, 282: multiple mentions of increase in pH due to carbonates. However, in Table 4, the measurement campaigns with higher concentrations of particulate carbonate (CaCO3) do not have higher pH values. Therefore, I do not understand the emphasis on this point, as it was not observed in the data.

**Response:**

Dependencies between the river pH and the carbonate system are complex. In the campaigns mentioned, we see high concentrations of DOC and of CaCO3. On the one hand, high concentrations of DOC indicate high decomposition rates and thus an input of CO2 to the rivers. Such a CO2 input would lower the water pH due to a change in the distribution of dissolved inorganic carbon (DIC). On the other hand, high concentrations of CaCO3 indicate an input of dissolved carbonates to the river. Such an increase in carbonates would shift the distribution of DIC away from CO2 and cause a pH increase.

Both of these processes occur simultaneously in the Simunjan river. The overall decrease in pH indicates that the impact of the CO2 input is dominant. However, the concurrent carbonate input likely buffered the pH decrease. This indicates that without the high carbonate concentrations, the river pH would have decreased even stronger, which would have decreased the in-river decomposition due to the limiting pH dependency.

**Referee:**

- why do you think you captured this in Simunjan but not other sites?

**Response:**

We can only speculate about the source of increased DOC and CaCO3 concentrations. Sources we suggest in the manuscript are:

[...] increased erosion of mineral soils due to deforestation in mountain regions upstream or liming practices in plantations along the river [...]

**Referee:**

- ideas for future work/speculation- if relevant, could you discuss the transition in pH as water flows from peatland drainage canals to streams to large rivers - where do you expect the pH constraint to be lifted? And could that help predict hotspots of CO2 emissions that would warrant further investigation?

**Response:**

The highest carbon (and CO2) concentrations can be found in small river arms with high peat coverage. However, these are also the river parts that exhibit the lowest water pH. Thus, decomposition rates (CO2 production per available DOC) are the lowest in these regions of high DOC. This natural distribution limits the CO2 production in river sections of high carbon content. When the water discharges to larger parts of the river, the water pH will increase, causing higher decomposition rates. However, the DOC concentration in these larger river sections is significantly lower, decreasing the absolute decomposition.

**Referee:**

-Can you please discuss further how you account for spatial and temporal variability? Fig 1- How does the distance upriver and sampling season influence the concentrations measured? How did you handle this? How might limitations related to number of sampling times and locations influence your results?

**Response:**

We included a discussion of this in the appendix B. In appendix B1 we discuss the impact of sampling locations. We state:

The data for this study was collected from samples taken in river sections that flow through peat soil. This ensures that the impact of peat soils on the river parameters is captured.

Concentrations measured in the small Malaysian rivers (Maludam and Sebuyau and Simunjan), with the exception of the Simunjan campaigns in Jamuary 2016 and March 2017 (Tab. 4, Fig. B3), show little variation over the river path and between campaigns (Fig. B1, B2 B3). However, the larger rivers drain mineral soils for the majority of their path and only reach peat regions close to the coast. Those rivers exhibit stronger differences in carbon concentrations along the length of the river. Rixen et al. (2010) found that DOC concentrations in the Siak river are by a factor of up to 4 higher in coastal peat regions than in the upstream river. CO2 concentrations in the large Sumatran rivers were not measured outside of the coastal peat regions. Due to the lower pH in river parts that cut through peat and the related pH limitation of DOC decomposition, the difference in CO2 concentrations along the river is likely lower than the difference in DOC concentrations. This is also indicated by CO2 measurements in the Rajang River that reveal CO2 concentrations in the peat-draining rivers sections to be only (15 – 20) % higher than CO2 concentrations upstream the peat regions (Müller-Dum et al., 2019).

The impact of the sampling time is discussed in appendix B2. We state:

The Southeast Asian study area is impacted by the Malaysian-Australian monsoon that causes presence of moisture loaded air with high precipitation rates from October to April while dry air dominates from May to September. To catch the impact of these rain and dry seasons on river carbon dynamics, campaigns in different months of the year were performed (Tab. A1). Yet, the seasonal data coverage is not dense enough to clearly identify or disprove a seasonal pattern in the measured data (Fig. B4, B5 & B6).

-----Data availability- The work of compiling the large dataset presented here is a major contribution to the community. Where will this dataset be made available for future research?

**Response:**

Averaged data per campaign are included in the Supplementary data. Raw data of individual sampling stations at the Malaysian rivers can be requested from us at the IUP Bremen.

**Referee:**

-Code availability – can you make any of the analyses you completed public or visible?

**Response:**

We are happy to provide the Python code used to derive the least-squares analysis within the Supplementary material. We will mention this in the Supplementary section.

**Referee:**

-----Other comments:

-Ln 33-39: awkward phrasing, hard to understand meaning without reading multiple times

**Response:**

We rephrased these lines and changed them to:

[...] Different reasons for this were suggested in literature. Müller et al. (2015) suggested short residence times of peat derived DOC in rivers due to the location of peatlands near the coast as a possible cause. Other suggestions are the recalcitrant nature of DOC (Müller et al., 2016) and the lack of oxygen (O2, Wit et al., 2015) which both lower the rate of DOC decomposition. Moreover, Borges et al. (2015) suggested a limitation of bacterial production and the resulting DOC decomposition in African peatdraining rivers as a consequence of low pH based on observations at rivers in the Congo basin. [...]

**Referee:**

-Ln 56: The Methods section would strongly benefit from a short "Overview" or "roadmap" paragraph near the beginning of the section. It is currently very difficult to follow the overall approach, and requires reading multiple times.

**Response:**

The methods section was restructured into two distinct parts that are introduced by a short overview paragraph:

This study's methods were separated into two parts. The first part provides information on the study area, conducted measurement campaigns and collected data that our analyses are based on. The second part describes the processes and equations used to quantify the decomposition dependency on O2 and pH.

**Referee:**

-Ln 19: "potential hotspot"?

**Response:**

Yes, we changed the phrasing accordingly.

-Figure 1: Can you mark the location of sampling?

**Response:**

The Sampling locations for the individual campaigns were included to the map as suggested.

**Referee:**

-Ln 149: You mention many data sources here – please reference. Are these from others or this work? Citations or reference to data in Supplement?

**Response:**

We included the according references in the text:

[...] The  $k_{600}$  we list in this study are based on a variety of techniques, including floating chamber measurements (Müller et al., 2015), calculations based on wind speed and catchment parameters (Müller-Dum et al., 2019) and balance models of water parameters (Rixen et al., 2008). [...]

**Referee:**

-Key points on the pH limitation vs. oxygen limitation are buried within the conclusion. Perhaps you could use sub-headers or more active topic sentence to make sure that the central points are clearly communicated.

**Response:**

We restructured the conclusions. The revised conclusions start with a paragraph about the data correlation and observed CO2 stagnation:

Our study shows that CO2 concentrations in and emissions from Southeast Asian rivers stagnate for high peat coverages of the river catchments. Despite further increase in river DOC concentrations, CO2 concentrations are fairly constant for peat coverages > 50 %. We find that this stagnation is caused by a natural limitation of DOC decomposition in these rivers. This process provides an answer to the question of why CO2 emissions from tropical peat-draining rivers are more moderate.

This is followed by a paragraph about the derived O2 and pH limitations:

Correlation to measured data indicates that the limitation in decomposition is mainly caused by low river pH. Data reveal an exponential limitation of DOC decomposition by pH as the most realistic scenario. This reduces the CO2 production in rivers of high peat coverage by up to 85 %. The limiting impact of O2 on decomposition in the rivers results to be comparatively small with